# Compression based bound for non-compressed network: unified generalization error analysis of large compressible deep neural network

**Taiji Suzuki**
Graduate School of Information Science and Technology, The University of Tokyo, Japan
Center for Advanced Intelligence Project, RIKEN, Japan
Japan Digital Design
taiji@mist.i.u-tokyo.ac.jp

**Hiroshi Abe**
iPride Co., Ltd., Japan,
abe@ipride.co.jp,

**Tomoaki Nishimura**
NTT Data Corporation, Japan,
Tomoaki.Nishimura@nttdata.com

## Abstract

One of the biggest issues in deep learning theory is the generalization ability of networks with huge model size. The classical learning theory suggests that overparameterized models cause overfitting. However, practically used large deep models avoid overfitting, which is not well explained by the classical approaches. To resolve this issue, several attempts have been made. Among them, the compression based bound is one of the promising approaches. However, the compression based bound can be applied only to a compressed network, and it is not applicable to the non-compressed original network. In this paper, we give a unified framework that can convert compression based bounds to those for non-compressed original networks. The bound gives even better rate than the one for the compressed network by improving the bias term. By establishing the unified framework, we can obtain a data dependent generalization error bound which gives a tighter evaluation than the data independent ones.

## 1 Introduction

Deep learning has shown quite successful results in wide range of machine learning applications. such as image recognition (Krizhevsky et al., 2012), natural language processing (Devlin et al., 2018) and image synthesis tasks (Radford et al., 2015). The success of deep learning methods is mainly due to its flexibility, expression power and computational efficiency for large dataset training. Due to its significant importance in wide range of application areas, its theoretical analysis is also getting much important. For example, it has been known that the deep neural network has universal approximation capability (Cybenko, 1989; Hornik, 1991; Sonoda & Murata, 2015) and its expressive power grows up in an exponential order against the number of layers (Montufar et al., 2014; Bianchini & Scarselli, 2014; Cohen et al., 2016; Cohen & Shashua, 2016; Poole et al., 2016; Suzuki, 2019). However, theoretical understandings are still lacking in several important issues.

Among several topics of deep learning theories, a generalization error analysis is one of the biggest issues in the machine learning literature. An important property of deep learning is that it generalizes well even though its parameter size is quite large compared with the sample size (Neyshabur et al., 2019). This can not be well explained by a classical VC-dimension type theory (Harvey et al., 2017) which suggests that overparameterized models cause overfitting and thus result in poor generalization ability.

For this purpose, norm based bounds have been extensively studied so far (Neyshabur et al., 2015; Bartlett et al., 2017b; Neyshabur et al., 2017; Golowich et al., 2018). These bounds are beneficial

because the bounds are not explicitly dependent on the number of parameters and thus are useful to explain the generalization error of overparameterized network (Bartlett, 1998; Neyshabur et al., 2015; 2019). However, these bounds are typically exponentially dependent on the number of layers and thus tends to be loose for deep network situations (Dziugaite & Roy, 2017; Arora et al., 2018; Nagarajan & Kolter, 2019). As a result, Arora et al. (2018) reported that a simple VC-dimension bound (Li et al., 2018; Harvey et al., 2017) can still give sharper evaluations than these norm based bounds in some practically used deep networks. Wei & Ma (2019) improved this issue by involving a data dependent Lipschitz constant as performed in Arora et al. (2018); Nagarajan & Kolter (2019).

On the other hand, *compression based bound* is another promising approach for tight generalization error evaluation which can avoid the exponential dependence on the depth. The complexity of deep neural network model is regulated from several aspects. For example, we usually impose explicit regularization such as weight decay (Krogh & Hertz, 1992), dropout (Srivastava et al., 2014; Wager et al., 2013), batch-normalization (Ioffe & Szegedy, 2015), and mix-up (Zhang et al., 2018; Verma et al., 2018). Zhang et al. (2016) reported that such explicit regularization does not have much effect but implicit regularization induced by SGD (Hardt et al., 2016; Gunasekar et al., 2018; Ji & Telgarsky, 2019) is important. Through these explicit and implicit regularizations, deep learning tends to produce a simpler model than its full expression ability (Valle-Perez et al., 2019; Verma et al., 2018). To measure how "simple" the trained model is, one of the most promising approaches currently investigated is the compression bounds (Arora et al., 2018; Baykal et al., 2019; Suzuki et al., 2018). These bounds measure how much the network can be compressed and characterize the size of the compressed network as the implicit effective dimensionality. Arora et al. (2018) characterized the implicit dimensionality based on so called *layer-cushion* quantity and suggested to perform random projection to obtain a compressed network. Along with a similar direction, Baykal et al. (2019) proposed a pruning scheme called Corenet and derived a bound of the size of the compressed network. Suzuki et al. (2018) has developed a spectrum based bound for their compression scheme. Unfortunately, all of these bounds guarantee the generalization error of only the compressed network, not the original network. Hence, it does not give precise explanations about why large network can avoid overfitting.

In this paper, we derive a unified framework to obtain a compression based bound for a *non-compressed network*. Unlike the existing researches, our bound is valid to evaluate the original network *before* compression, and thus gives a direct explanation about why deep learning generalizes despite its large network size. The difficulty to apply the compression bound to the original network lies in evaluation of the population $L_2$-bound between the compression network and the original network. A naive evaluation results in the VC-bound which is not preferable. This difficulty is overcome by developing novel data dependent capacity control technique using *local Rademacher complexity* bounds (Mendelson, 2002; Bartlett et al., 2005; Koltchinskii, 2006; Giné & Koltchinskii, 2006). Then, the bound is applied to some typical situations where the network is well compressed. Our analysis stands on the implicit bias hypothesis (Gunasekar et al., 2018; Ji & Telgarsky, 2019) that claims deep learning tends to produce rather simple models. Actually, Gunasekar et al. (2018); Ji & Telgarsky (2019) showed gradient descent results in (near) low rank parameter matrices in each layer in linear network settings. Martin & Mahoney (2018) evaluated the eigenvalue decays of the weight matrix through random matrix theories and several numerical experiments. These observations are also supported by the *flat minimum* analysis (Hochreiter & Schmidhuber, 1997; Wu et al., 2017; Langford & Caruana, 2002), that is, the product of the eigenvalues of the Hessian around the SGD solution tends to be small, which means SGD converges to a flat minimum and possess stability against small perturbations leading to good generalization. Based on these observations, we make use of the eigenvalue decay of the weight matrix and the covariance matrix among the nodes in each layer (this assumption is actually verified by numerical experiments in Appendix D). The eigenvalue decay speed characterizes the redundancy in each layer and thus is directly relevant to compression ability. Our contributions in this paper are summarized as follows:

- We give a unified framework to obtain a compression based bound for *non-compressed* network which properly explains that a compressible network can generalizes well. The bound can convert several existing compression based bounds to that for non-compressed one in a unifying manner. The bound is applied to near low rank models as concrete examples.

- We develop a data dependent capacity control technique to bound the discrepancy between the original network and compressed network. As a result, we obtain a sharp generalization

Table 1: Comparison of each generalization error to our bound. $R_F$ is the Frobenius norm of the weight matrix, $R_2$ is the operator norm of the weight matrix, $R_{p\to q}$ is the $(p,q)$ matrix norm, $L$ is the depth, $m$ is the maximum of the width, $n$ is the sample size. $\bar{R}_n$ and $\dot{R}_r$ represent the Rademacher complexity and local Rademacher complexity respectively. $\kappa$ is a Lipschitz constant between layers. $\alpha$ represents the eigenvalue drop rate of the weight matrix, and $\beta$ represents that of the covariance matrix among the nodes in each internal layer. $\hat{r}$ is the bias induced by compression. "Original" indicates whether the bound is about the original network or not.

| Authors | Rate | Bound type | Original |
|---|---|---|---|
| Neyshabur et al. (2015) | $\dfrac{2^L R_F^L}{\sqrt{n}}$ | Norm base | Yes |
| Bartlett et al. (2017b) | $\dfrac{R_2^L}{\sqrt{n}}\left(L\,\dfrac{R_{2\to1}^{2/3}}{R_2^{2/3}}\right)^{3/2}$ | Norm base | Yes |
| Wei & Ma (2019) | $\dfrac{\left(1+L\kappa^{\frac{4}{3}}R_{2\to1}^{2/3}+L\kappa^{\frac{2}{3}}R_{1\to1}^{2/3}\right)^{3/2}}{\sqrt{n}}$ | Norm base | Yes |
| Neyshabur et al. (2017) | $\dfrac{R_2^L}{\sqrt{n}}\sqrt{L^3 m\,\dfrac{R_F^2}{R_2^2}}$ | Norm base | Yes |
| Golowich et al. (2018) | $R_F^L \min\left\{\dfrac{1}{n^{1/4}},\sqrt{\dfrac{L}{n}}\right\}$ | Norm base | Yes |
| Li et al. (2018) Harvey et al. (2017) | $\dfrac{R_2^L\sqrt{L^2 m^2}}{\sqrt{n}}$ | VC-dim. | Yes |
| Arora et al. (2018) | $\hat{r}+\sqrt{\dfrac{L^2 \max\limits_{1\le i\le n}|\hat{f}(x_i)|^2 \sum_{\ell=1}^L \frac{1}{\mu_\ell^2 \mu_{\ell\to}^2}}{n\hat{r}^2}}$ | Compression | No |
| Suzuki et al. (2018) | $\hat{r}+\sqrt{\dfrac{\sum_{\ell=1}^L m_{\ell+1}^\sharp m_\ell^\sharp}{n}}$ | Compression | No |
| Ours (Thm. 1) | $\hat{r}\sqrt{\dfrac{1}{n}}+\dot{R}_{\hat{r}}(\widehat{\mathcal{F}}-\widehat{\mathcal{G}})+\bar{R}_n(\widehat{\mathcal{G}})$ | General | Yes |
| Ours (Cor. 1) | $\sqrt{L(L\kappa^2)^{1/\alpha}\dfrac{Lm}{n}}$ | Low rank weight | Yes |
| Ours (Thm. 4) | $\sqrt{\dfrac{L^{1+\frac{\beta}{(2\alpha-1)}+\beta}\,(Lm)^{\frac{4/\beta}{4/\beta+2(1-1/2\alpha)}}}{n}}$ | Low rank weight Low rank cov. | Yes |

error bound which is even better than that of the compressed network. All derived bounds are characterized by data dependent quantities.

**Other related work**   Recently, the role of over-parameterization for two layer networks has been extensively studied (Neyshabur et al., 2019; Arora et al., 2019). These are for the shallow network and the generalization error is essentially given by the norm based bounds. It is not obvious that these bounds also give sharp bounds for deep models.

PAC-Bayes bound is also applied to obtain a non-vacuous compression based bound (Zhou et al., 2019). However, the bound is still for the compressed (quantized) models and it is not obvious that that bound can be converted to that for the original network.

Relation between compression and learnability was traditionally studied in a different framework as in Littlestone & Warmuth (1986) and minimum description code length (Hinton & Van Camp, 1993). Our bound would share the same spirits with these studies but give a new analysis by incorporating recent observations in deep learning researches.

## 2   PRELIMINARIES: PROBLEM FORMULATION AND NOTATIONS

In this section, we give the problem setting and notations that will be used in the theoretical analysis. We consider the standard supervised leaning formulation where data consists of input $x \in \mathbb{R}^d$ and output (or label) $y \in \mathbb{R}$. We consider a single output setting, i.e., the output $y$ is a 1-dimensional real value, but it is straight forward to generalize the result to a multiple output case. Suppose that we are given $n$ i.i.d. observations $D_n = (x_i, y_i)_{i=1}^n$ distributed from a probability distribution $P$. To measure the performance of a trained function $\hat{f}$, we use a loss function $\psi : \mathbb{R} \times \mathbb{R} \to \mathbb{R}$ and define

a training error and its expected one as

$$\widehat{\Psi}(f) := \frac{1}{n} \sum_{i=1}^{n} \psi(y_i, f(x_i)), \;\; \Psi(f) := \mathrm{E}[\psi(Y, f(X))],$$

where the expectation is taken with respect to $(X, Y) \sim P$. Basically, we are interested in the generalization error $\Psi(\widehat{f}) - \widehat{\Psi}(\widehat{f})$ for an estimator $\widehat{f}$. We denote the empirical $L_2$-norm by $\|f\|_n := \sqrt{\sum_{i=1}^{n} f(z_i)^2/n}$ for an empirical observation $z_i = (x_i, y_i)$ $(i = 1, \dots, n)$. The population $L_2$-norm is denoted by $\|f\|_{L_2} := \sqrt{\mathrm{E}_{Z \sim P}[f(Z)^2]}$.

This paper deals with deep neural networks as a model. The activation function is denoted by $\eta$ which will be assumed to be 1-Lipschitz as satisfied by ReLU (Assumption 1). Let the depth of the network be $L$ and the width of the $\ell$-th internal layer be $m_\ell$ $(\ell = 1, \dots, L+1)$ where we set $m_1 = d$ (dimension of input) and $m_{L+1} = 1$ (dimension of output) for convention. Then, the set of networks having depth $L$ and width $\mathbf{m} = (m_1, \dots, m_L)$ with norm constraint as

$$\mathrm{NN}(\mathbf{m}, R_2', R_\mathrm{F}') := \Big\{ f(x) = G \circ (W^{(L)}\eta(\cdot)) \circ (W^{(L-1)}\eta(\cdot)) \circ \cdots \circ (W^{(1)}x) \;|$$
$$W^{(\ell)} \in \mathbb{R}^{m_\ell \times m_{\ell+1}}, \|W^{(\ell)}\|_2 \leq R_2', \|W^{(\ell)}\|_\mathrm{F} \leq R_\mathrm{F}' \Big\}.$$

where $\|W\|_2 := \sup_{u:\|Wu\|\neq 0} \|Wu\|/\|u\|$[1] is the operator norm (the maximum singular value), $\|W\|_\mathrm{F} := \sqrt{\sum_{i,j} W_{i,j}^2}$ is the Frobenius norm, and $G$ is the "clipping" operator that is defined by $G(x) = \max\{-M, \min\{x, M\}\}$ for a constant $M$. The reason why we put the clipping operator $G$ on top of the last layer is because the clipping operator restricts the $L_\infty$-norm by a constant $M$ and then we can avoid unrealistically loose generalization error. Note that the clipping operator does not change the classification error for binary classification. We express $\mathcal{F}$ to represent the "full model": $\mathcal{F} = \mathrm{NN}(\mathbf{m}, R_2, R_\mathrm{F})$ for a given $R_2, R_\mathrm{F} > 0$. Here, we implicitly suppose that $R_2$ is close to 1 so that the norm of the output from internal layers is not too much amplified, while $R_\mathrm{F}$ could be moderately large.

The *Rademacher complexity* is the typical tool to evaluate the generalization error on a function class $\mathcal{F}'$, which is denoted by $\hat{R}_n(\mathcal{F}') := \mathrm{E}_\epsilon \big[ \sup_{f \in \mathcal{F}'} \frac{1}{n} \sum_{i=1}^{n} \epsilon_i f(z_i) \mid D_n \big]$ where $D_n = (z_i)_{i=1}^{n} = (x_i, y_i)_{i=1}^{n}$, and $\epsilon_i$ $(i = 1, \dots, n)$ is an i.i.d. Rademacher sequence $(P(\epsilon_i = 1) = P(\epsilon_i = -1) = 1/2)$. This is also called conditional Rademacher complexity because the expectation is taken conditioned on fixed $D_n$. Its expectation with respect to $D_n$ is denoted by $\bar{R}_n(\mathcal{F}') := \mathrm{E}_{D_n}[\hat{R}_n(\mathcal{F}')]$. Roughly speaking the Rademacher complexity measures the size of the model and it gives an upper bound of the generalization error (Vapnik, 1998; Mohri et al., 2012).

The main difficulty in generalization error analysis of deep learning is that the Rademacher complexity of the full model $\mathcal{F}$ is quite large. One of the successful approaches to avoid this difficulty is the compression based bound (Arora et al., 2018; Baykal et al., 2019; Suzuki et al., 2018) which measures how much the trained network $\widehat{f}$ can be compressed. If the network can be compressed to much smaller one, then its intrinsic dimensionality can be regarded as small. To describe it more precisely, suppose that the trained network $\widehat{f}$ is included in a subset of the neural network model: $\widehat{f} \in \widehat{\mathcal{F}} \subset \mathcal{F}$. For example, $\widehat{\mathcal{F}}$ can be a set of networks with weight matrices that have bounded norms and are near low rank (Sec. 3.1 or Sec. 3.2). We do not assume a specific type of training procedure, but we give a uniform bound valid for any estimator $\widehat{f}$ that falls into $\widehat{\mathcal{F}}$ and satisfies the following compressibility condition. We suppose that the network $\widehat{f}$ is easy to compress, that is, $\widehat{f}$ can be compressed to a smaller network $\widehat{g}$ which is included in a submodel: $\widehat{g} \in \widehat{\mathcal{G}}$. For example, $\widehat{\mathcal{G}}$ can be a set of networks with a smaller size than $\widehat{f}$. How small the trained network $\widehat{f}$ can be compressed has been characterized by several notions such as "layer-cushion" (Arora et al., 2018). Typical compression based bounds give generalization errors of the compressed model $\widehat{g}$, not the original network $\widehat{f}$. Our approach converts an error bound of $\widehat{g}$ to that of $\widehat{f}$ and eventually obtains a tighter evaluation.

---

[1]In this paper, $\|\cdot\|$ denotes the Euclidean norm: $\|u\| = \sqrt{\sum_i u_i^2}$.

The biggest difficulty for transforming the compression bound to that of $\widehat{f}$ lies in evaluation of the population $L_2$-norm between $\widehat{f}$ and $\widehat{g}$. Basically, the compression based bounds are given as

$$\Psi(\widehat{g}) \leq \widehat{\Psi}(\widehat{f}) + \|\widehat{f} - \widehat{g}\|_n + C\bar{R}_n(\widehat{\mathcal{G}}), \tag{1}$$

for a constant $C > 0$ under some assumptions (Table 1). The term $\|\widehat{f} - \widehat{g}\|_n$ appears to adapt the empirical error of $\widehat{f}$ to that of $\widehat{g}$, that is called "compression error" which can be seen as a bias term. We see that, in the right hand side, there appears the complexity of $\widehat{\mathcal{G}}$ which is assumed to be much smaller than that of the full model $\mathcal{F}$. However, the left hand side is not the expected error of $\widehat{f}$ but that of $\widehat{g}$. One way to transfer this bound to that of $\widehat{f}$ is that we have $|\Psi(\widehat{g}) - \Psi(\widehat{f})| \leq \|\widehat{g} - \widehat{f}\|_{L_2}$ by assuming Lipschitz continuity of the loss function and then convert the bound (1) to

$$\Psi(\widehat{f}) \leq \widehat{\Psi}(\widehat{f}) + (\|\widehat{f} - \widehat{g}\|_n + \|\widehat{f} - \widehat{g}\|_{L_2}) + \bar{R}_n(\widehat{\mathcal{G}}).$$

However, to bound the term $\|\widehat{f} - \widehat{g}\|_n + \|\widehat{f} - \widehat{g}\|_{L_2}$, there typically appears the complexity of the model $\widehat{\mathcal{F}}$ which is larger than the compressed model $\widehat{\mathcal{G}}$ like $\|\widehat{f} - \widehat{g}\|_n \leq \sqrt{\|\widehat{f} - \widehat{g}\|_{L_2}^2 + O_p(\bar{R}(\widehat{\mathcal{F}}))}$, which results in slow convergence rate. To overcome this difficulty, we need to carefully control the difference between the training and test error of $\widehat{f}$ and $\widehat{g}$ by utilizing the local Rademacher complexity technique (Mendelson, 2002; Bartlett et al., 2005; Koltchinskii, 2006; Giné & Koltchinskii, 2006). The local Rademacher complexity of a model $\mathcal{F}'$ with radius $r > 0$ is defined as

$$\dot{R}_r(\mathcal{F}') := \bar{R}_n(\{f \in \mathcal{F}' \mid \|f\|_{L_2} \leq r\}).$$

The main difference from the standard Rademacher complexity is that the model is localized to a set of functions satisfying $\|f\|_{L_2} \leq r$. As a result, we obtain a tighter error bound.

Throughout this paper, we always assume the following assumptions. Let $P_{\mathcal{X}}$ and $P_{\mathcal{Y}}$ denote the marginal distribution of $x$ and that of $y$ respectively.

**Assumption 1** (Lipschitz continuity of loss and activation functions). *The loss function $\psi$ is 1-Lipschitz continuous with respect to the function output:*

$$|\psi(y, u) - \psi(y, u')| \leq |u - u'| \ \ (\forall y \in \mathrm{supp}(P_{\mathcal{Y}}), \ u, u' \in \mathbb{R}).$$

*The activation function $\eta$ is also 1-Lipschitz continuous: $\|\eta(u) - \eta(u')\| \leq \|u - u'\| \ (\forall u \in \mathbb{R}^{d'})$ where $d'$ is any positive integer.*

**Assumption 2.** *The norm of input is bounded by $B_x > 0$: $\|x\| \leq B_x \ \ (\forall x \in \mathrm{supp}(P_{\mathcal{X}}))$.*

**Assumption 3.** *The $L_\infty$-norms of all elements in $\widehat{\mathcal{F}}$ and $\widehat{\mathcal{G}}$ are bounded by $M \geq 1$: $\|f\|_\infty, \|g\|_\infty \leq M$ for all $f \in \widehat{\mathcal{F}}$ and $g \in \widehat{\mathcal{G}}$.*

This assumption can be ensured by applying the clipping operator $G$ to the output of the functions. In this paper, all the variables $L, m_\ell, R_2, R_F, M, B_x$ are supposed to be $o(n)$. What we will derive in the following is a bound which has mild dependency on the depth $L$ and depends on the width $(m_\ell)_{\ell=1}^L$ in a sub-linear order by using the compression based approach.

**Existing bounds for no-compressed network**  Here we give a brief review of the generalization error bound for non-compressed models. (i) VC-bound: The Rademacher complexity of the full model $\mathcal{F}$ can be bounded by a naive VC-dimension bound (Harvey et al., 2017) which is $\bar{R}_n(\mathcal{F}) = O\left(\sqrt{\frac{L\sum_{\ell=1}^n m_\ell m_{\ell+1}}{n}\log(n)}\right)$. In this bound, there appears the number of parameters $\sum_{\ell=1}^n m_\ell m_{\ell+1}$ in the numerator. However, the number of parameters is often larger than the sample size $n$ in practical use. Hence, this bound is not appropriate to evaluate generalization ability of overparameterized networks. (ii) Norm-based bound: Golowich et al. (2018) showed the norm based bound which is given as $\bar{R}_n(\mathcal{F}) = O\left(\sqrt{\frac{LR_F^L}{n}}\right)$. However, this is exponentially dependent on the depth as $R_F^L$ resulting in quite loose bound. Neyshabur et al. (2017) showed a norm based bound of $\bar{R}_n(\mathcal{F}) = O\left(\sqrt{\frac{L^3(\max_\ell m_\ell)R_F^2/R_2^2}{n}}\right)$ which avoids the exponential dependency. However, there is still dependency on the width, $\sum_\ell m_\ell R_F^2$, which is larger than the

linear order of the width since $R_F$ could be moderately large. Bartlett et al. (2017b) showed $\bar{R}_n(\mathcal{F}) = O\left(\frac{R_2^L}{\sqrt{n}}\left(L\frac{R_{2\to 1}^{2/3}}{R_2^{2/3}}\right)^{3/2}\right)$. The norm constraint on $R_{2\to 1}$ implicitly assumes sparsity on the weight matrix and $R_{2\to 1}$ typically depends on the width linearly. Wei & Ma (2019) improved the exponential dependency $R_2^L$ appearing in this bound (Bartlett et al., 2017b) to obtained a bound $O\left(\frac{1}{\sqrt{n}}\left(1 + L\kappa^{\frac{4}{3}}R_{2\to 1}^{2/3} + L\kappa^{\frac{2}{3}}R_{1\to 1}^{2/3}\right)^{3/2}\right)$ where $\kappa$ is the Lipschitz continuity between layers. We can see that $R_{2\to 1}^2$ and $R_{1\to 1}^2$ can depend on the width linearly and quadratically respectively even though $R_2$ is bounded.

## 3 COMPRESSION BOUND FOR NONCOMPRESSED NETWORK

Here, we give a general theoretical tool that converts a compression based bound to that for the original network $\widehat{f}$. We suppose the model classes $\widehat{\mathcal{F}}$ and $\widehat{\mathcal{G}}$ are fixed independently on each data observation[2]. For sets of functions, $\mathcal{F}'$ and $\mathcal{G}'$, we denote the Minkowski difference of them by $\mathcal{F}' - \mathcal{G}' := \{f - g \mid f \in \mathcal{F}', g \in \mathcal{G}'\}$. We assume that the local Rademacher complexity of $\widehat{\mathcal{F}} - \widehat{\mathcal{G}}$ has a concave shape with respect to $r > 0$: Suppose that there exists a function $\phi : [0, \infty) \to [0, \infty)$ such that

$$\dot{R}_r(\widehat{\mathcal{F}} - \widehat{\mathcal{G}}) \le \phi(r) \text{ and } \phi(2r) \le 2\phi(r) \ (\forall r > 0).$$

This condition is not restrictive, and usual bounds for the local Rademacher complexity satisfy this condition (Mendelson, 2002; Bartlett et al., 2005). Using this notation, we define $r_* = r_*(t)$ as

$$r_*(t) := \inf\left\{r > 0 \,\middle|\, 8\frac{\phi(r)}{r^2} + M\sqrt{\frac{4t}{r^2 n}} + M^2\frac{2t}{r^2 n} \le \frac{1}{2}\right\}. \tag{2}$$

This is roughly given by the fixed point of a function $r^2 \mapsto \phi(r)$, and it is useful to bound the *ratio* of the empirical $L_2$-norm and the population $L_2$-norm of an element $h$ in $\widehat{\mathcal{F}} - \widehat{\mathcal{G}}$: $\|h\|_{L_2}^2 / (\|h\|_n^2 + r_*^2) \le 1/2$ with high probability. Finally, we denote $\psi(\mathcal{F}') := \{\psi(\cdot, f(\cdot)) \mid f \in \mathcal{F}'\}$ for a set $\mathcal{F}'$ of functions. Then, we obtain the following theorem that gives the compression based bound for non-compressed networks.

**Theorem 1.** *Suppose that the empirical $L_2$-distance between $\widehat{f}$ and $\widehat{g}$ is bounded by $\|\widehat{f} - \widehat{g}\|_n \le \hat{r}^2$ for a fixed $\hat{r} > 0$ almost surely. Let $\dot{r} := \sqrt{2(\hat{r}^2 + r_*^2)}$, then, under Assumptions 1, 2, 3, there exists a universal constant $C > 0$ such that*

$$\Psi(\widehat{f}) \le \widehat{\Psi}(\widehat{f}) + \underbrace{2\bar{R}_n(\widehat{\mathcal{G}}) + \sqrt{M\frac{2t}{n}}}_{\text{main term}} + C\underbrace{\left[\dot{R}_{\dot{r}}(\psi(\widehat{\mathcal{F}}) - \psi(\widehat{\mathcal{G}})) + \dot{r}\sqrt{\frac{t}{n}} + \frac{1 + tM}{n}\right]}_{\text{bias term}}.$$

*with probability at least $1 - 3e^{-t}$ for all $t \ge 1$.*

The proof is given in Appendix A. The bound consists of two terms: "main term" and "bias term." The main term represents the complexity of the compressed model $\widehat{\mathcal{G}}$ which could be much smaller than $\widehat{\mathcal{F}}$. The bias term represents a sample complexity to bridge the original model and the compressed model. Typically we have $r_*^2 = o(1/\sqrt{n})$, and if we set $\hat{r} = o_p(1)$, then the bias term can be faster than the main term which is $O(1/\sqrt{n})$. The term $\dot{R}_{\dot{r}}(\psi(\widehat{\mathcal{F}}) - \psi(\widehat{\mathcal{G}}))$ can be refined a little bit and the refined term can be evaluated by using the *covering number* of the model. The refined version is given in Appendix A (Theorem 5). This bound is general, and can be combined with the compression bounds derived so far such as Arora et al. (2018); Baykal et al. (2019); Suzuki et al. (2018) where the complexity of $\widehat{\mathcal{G}}$ and the bias $\hat{r}$ are analyzed for their generalization error bounds.

The main difference from the compression bound (1) for $\widehat{g}$ is that the bias term $\hat{r} = \|\widehat{f} - \widehat{g}\|_n$ is replaced by $\frac{1}{\sqrt{n}}\|\widehat{f} - \widehat{g}\|_n$ which is $\sqrt{n}$ times smaller. Since $r_*^2$ is typically $o(1/\sqrt{n})$ and $\dot{R}_{\dot{r}}(\psi(\widehat{\mathcal{F}}) -$

---

[2] We can extend the result to data dependent models $\widehat{\mathcal{F}}$ and $\widehat{\mathcal{G}}$ by taking uniform bound for all possible choice of the pair $\widehat{\mathcal{F}}$ and $\widehat{\mathcal{G}}$. However, we omit explicit presentation of this uniform bound for simplicity.

$\psi(\widehat{\mathcal{G}})$) can be made in the same order as the main term or even faster by setting $\hat{r}$ appropriately, we may neglect these terms. Then, the bound is informally written as

$$\Psi(\widehat{f}) \leq \hat{\Psi}(\widehat{f}) + O_p\left(\bar{R}_n(\widehat{\mathcal{G}}) + \tfrac{1}{\sqrt{n}}\|\widehat{f} - \widehat{g}\|_n + \sqrt{1/n}\right).$$

This allows us to obtain tighter bound than the compression bound for $\widehat{g}$ because the bias term $\hat{r}/\sqrt{n}$ is much smaller than $\hat{r}$ and eventually we can let the variance term $\bar{R}_n(\widehat{\mathcal{G}})$ much smaller by taking small compressed model $\widehat{\mathcal{G}}$ when we balance the bias and variance trade-off. This is an advantageous point of directly bounding the generalization error of $\widehat{f}$ instead of $\widehat{g}$.

Finally, we note that some existing bounds such as Arora et al. (2018); Bartlett et al. (2017b); Wei & Ma (2019) assumes a constant margin so that the bias term can be a sufficiently small constant (which does not need to converge to 0). On the other hand, our bound does not assume it and the bias term should converge to 0 so that the bias is balanced with the variance term, which is a more difficult problem setting.

**Example 1.** *In practice, a trained network can be usually compressed to one with sparse weight matrix via pruning techniques (Denil et al., 2013; Denton et al., 2014). Based on this observation, Baykal et al. (2019) derived a compression based bound based on a pruning procedure. In this situation, we may suppose that $\widehat{\mathcal{G}}$ is the set of networks with $S$ non-zero parameters where $S$ is much smaller than the total number of parameters: $\widehat{\mathcal{G}} = \{f \in \mathrm{NN}(\mathbf{m}, R_2, R_\mathrm{F}) \mid \sum_{\ell=1}^{L} \|W^{(\ell)}\|_0 \leq S\}$ where $\|W^{(\ell)}\|_0$ is the number of nonzero parameters of the weight matrix $W^{(\ell)}$. In this situation, its Rademacher complexity is bounded by $\bar{R}(\widehat{\mathcal{G}}) \leq CM\sqrt{L\frac{S}{n}\log(n)}$ (see Appendix B.2 for the proof). This is much smaller than the VC-dimension bound $\sqrt{\frac{L\sum_{\ell=1}^{n} m_\ell m_{\ell+1}}{n}\log(n)}$ if $S \ll \sum_{\ell=1}^{n} m_\ell m_{\ell+1}$.*

Although our bound can be adopted to several compression based bounds, we are going to demonstrate how small the obtained bound can be through some typical situations in the following.

### 3.1 COMPRESSION BOUND WITH NEAR LOW RANK WEIGHT MATRIX

Here, we analyze the situation where the trained network has near low rank weight matrices $(W^{(\ell)})_{\ell=1}^{L}$. It has been reported that the trained network tends to have near low rank weight matrices experimentally (Gunasekar et al., 2018; Ji & Telgarsky, 2019) (see Appendix D for the empirical verification). This situation has been analyzed in Arora et al. (2018) where the low rank property is characterized by their original quantities such as layer cushion. However, we employ a much simpler and intuitive condition to highlight how the low rank property affects the generalization.

**Assumption 4.** *Assume that each of weight matrices $W^{(\ell)}$ ($\ell = 1, \ldots, L$) of any $f \in \widehat{\mathcal{F}}$ is near low rank, that is, there exists $\alpha > 1/2$ and $V_0 > 0$ such that*

$$\sigma_j(W^{(\ell)}) \leq V_0 j^{-\alpha},$$

*where $\sigma_j(W)$ is the $j$-th largest singular value of a matrix $W$ ($\sigma_1(W) \geq \sigma_2(W) \geq \cdots \geq 0$).*

In this situation, we can see that for any $1 \leq s \leq \min\{m_\ell, m_{\ell+1}\}$, we can approximate $W^{(\ell)}$ by a rank $s$ matrix $W'$ as $\|W^{(\ell)} - W'\|_2 \leq V_0 s^{-\alpha}$. Let the set of networks with exactly low rank weight matrices be $\mathrm{NN}(\mathbf{m}, \mathbf{s}, R_2, R_\mathrm{F}) := \{f \in \mathrm{NN}(\mathbf{m}, R_2, R_\mathrm{F}) \mid$ the weight matrix $W^{(\ell)}$ of $f$ has rank $s_\ell\}$ for $\mathbf{s} = (s_1, \ldots, s_L)$. If we set $\widehat{\mathcal{G}} = \mathrm{NN}(\mathbf{m}, \mathbf{s}, R_2, R_\mathrm{F})$, then we have the following theorem.

**Theorem 2.** *The compressed model $\widehat{\mathcal{G}} = \mathrm{NN}(\mathbf{m}, \mathbf{s}, R_2, R_\mathrm{F})$ has the following complexity:*

$$\bar{R}_n(\widehat{\mathcal{G}}) \leq CM\sqrt{L\frac{\sum_{\ell=1}^{L} s_\ell(m_\ell + m_{\ell+1})}{n}\log(n)}.$$

*If $\widehat{\mathcal{F}}$ satisfies Assumption 4, we can set $\hat{r} = V_0 R_2^{L-1} B_x \sum_{\ell=1}^{L} s_\ell^{-\alpha}$: for any $\widehat{f} \in \widehat{\mathcal{F}}$, there exists $\widehat{g} \in \widehat{\mathcal{G}}$ such that $\|\widehat{f} - \widehat{g}\|_n \leq \hat{r}$. Then, letting $A_1 = L\frac{\sum_{\ell=1}^{L} s_\ell(m_\ell + m_{\ell+1})}{n}\log(n)$ and*

$A_2 = L \frac{(\sum_{\ell=1}^{L} m_\ell)(2LV_0R_2^{L-1}B_x)^{1/\alpha}}{n}$, *the overall generalization error is bounded by*

$$\Psi(\widehat{f}) \leq \widehat{\Psi}(\widehat{f}) + C\left[MA_1 + M^{\frac{2\alpha-1}{2\alpha+1}}A_2^{\frac{2\alpha}{1+2\alpha}} + \sqrt{\widehat{r}^{2(1-2\alpha)}A_2} + (\widehat{r}+M)\sqrt{A_1} + \frac{1+tM}{n}\right],$$

*with probability* $1 - 3e^{-t}$ *for any* $t > 1$ *where* $C > 0$ *is a constant depending on* $\alpha$.

See Appendix B.3 for the proof. This indicates that, if $\alpha > 1/2$ is large (in other words, each weight matrix is close to rank 1), then we have a better generalization error bound. Note that the rank $s_\ell$ can be arbitrary chosen and $\hat{r}$ and $A_1$ are in a trade-off relation. Hence, by selecting the rank appropriately so that this trade-off is balanced, then we obtain the optimal upper bound as in the following corollary.

**Corollary 1.** *Under Assumption 4, using the same notation as Theorem 2, it holds that*

$$\Psi(\widehat{f}) \leq \widehat{\Psi}(\widehat{f}) + C\left[M^{1-1/2\alpha}\sqrt{L\frac{(\sum_{\ell=1}^{L} m_\ell)(2LV_0R_2^{L-1}B_x)^{1/\alpha}}{n}\log(n)} + M^{\frac{2\alpha-1}{2\alpha+1}}A_2^{\frac{2\alpha}{2\alpha+1}} + \frac{1+tM}{n}\right]$$

*with probability* $1 - 3e^{-t}$ *for any* $t > 1$ *where* $C$ *is a constant depending on* $\alpha$.

An important point here is that the bound is $O(\sqrt{L\frac{\sum_{\ell=1}^{L} m_\ell}{n}})$ which has linear dependency on the width $m_\ell$ in the square root, but the naive VC-dimension bound has quadratic dependency $O(\sqrt{L\frac{\sum_{\ell=1}^{L} m_\ell m_\ell}{n}})$. In other words, the term in the square root has linear dependency to the number of *nodes* instead of the number of *parameters*. This is huge gap because the width can be quite large in practice. This result implies that a compressible model achieves much better generalization than the naive VC-bound.

In the generalization error bound, there appears $R_2^L$. Even though $R_2$ can be much smaller than $R_F$, the exponential dependency $R_2^L$ can give loose bound as pointed out in Arora et al. (2018). This is due to a rough evaluation of the Lipschitz continuity between layers, but the practically observed Lipschitz constant is usually much smaller. To fix this issue, we give a refined version of Corollary 1 in Appendix B.4 by using data dependent Lipschitz constants such as interlayer cushion and interlayer smoothness introduced by Arora et al. (2018). The refined bound does not involve the exponential term $R_2^L$, but instead $\kappa^2$ ($\kappa$: Lipschitz continuity) appears.

### 3.2 COMPRESSION BOUND WITH NEAR LOW RANK COVARIANCE MATRIX

Strictly speaking, the near low rank condition on the weight matrix in the previous section can be dealt with a standard Rademacher complexity argument. Here, we consider more data dependent bound: We assume the near low rank property of the *covariance matrix* among the nodes in an internal layer (see Appendix D for the empirical verification). A compression based bound for $\widehat{g}$ using the low rank property of the covariance has been studied by Suzuki et al. (2018), but their analysis requires a bit strong condition on the weight matrix. In this paper, we employ a weaker assumption.

Let $\widehat{\Sigma}_{(\ell)} = \frac{1}{n}\sum_{i=1}^{n}\phi_\ell(x_i)\phi_\ell(x_i)^\top$ be the covariance matrix of the nodes in the $\ell$-th layer where $\phi_\ell(x) = \eta \circ (W^{(\ell-1)}\eta(\cdot)) \circ \cdots \circ (W^{(1)}x)$.

**Assumption 5.** *Suppose that the trained network* $\widehat{f}$ *satisfies the following conditions:*

$$\sigma_j(\widehat{\Sigma}_{(\ell)}) \leq \dot{\mu}_j^{(\ell)} =: U_0 j^{-\beta}, \tag{3}$$

*for a fixed* $\beta > 1$ *and* $U_0 > 0$.

If $\widehat{f}$ satisfies this assumption, then we can show that $\widehat{f}$ can be compressed to a smaller one $f^\sharp$ that has width $(m_\ell^\sharp)_{\ell=2}^{L}$ with compression error roughly evaluated as $\|\widehat{f} - f^\sharp\|_n \lesssim \sum_\ell (m_\ell^\sharp)^{-\beta/2}$. More precisely, for given $\tilde{r}_\ell > 0$ ($\ell = 1, \ldots, L$) which corresponds to the compression error in the $\ell$-th layer, let $\dot{m}_\ell := \max\{1 \leq j \leq m_\ell \mid \dot{\mu}_j^{(\ell)} \geq \tilde{r}_\ell^2/4\}$. Then, we define $N_\ell(\tilde{\mathbf{r}}) = \frac{\beta+1}{\beta-1}\dot{m}_\ell + 8\frac{(\sum_{k=1}^{\ell-1} R_2^{(\ell-1-k)}R_F\tilde{r}_k)^2}{\tilde{r}_\ell^2}$, for $\tilde{\mathbf{r}} = (\tilde{r}_1, \ldots, \tilde{r}_L)$. Correspondingly we set

$$m_\ell^\sharp := 5N_\ell(\tilde{\mathbf{r}})\log(80N_\ell(\tilde{\mathbf{r}})).$$

Then, we obtain the following theorem.

**Theorem 3.** *Let* $\hat{r} := \sum_{k=1}^{L} R_2^{(L-k)} R_{\mathrm{F}} \tilde{r}_k$. *Then, under Assumption 5, there exists* $\widehat{g}$ *with width* $\mathbf{m}^{\sharp} = (m_1, m_2^{\sharp}, \ldots, m_L^{\sharp})$ *that satisfies* $\widehat{g} \in \mathrm{NN}(\mathbf{m}^{\sharp}, \sqrt{\frac{20}{3} \max_{\ell} m_{\ell}} R_2 \sqrt{\frac{20}{3} \max_{\ell} m_{\ell}} R_{\mathrm{F}})$ *and*

$$\|\widehat{f} - \widehat{g}\|_n \leq \hat{r}.$$

*In particular, we may set* $\widehat{\mathcal{G}} = \mathrm{NN}(\mathbf{m}^{\sharp}, \sqrt{\frac{20}{3} \max_{\ell} m_{\ell}} R_2, \sqrt{\frac{20}{3} \max_{\ell} m_{\ell}} R_{\mathrm{F}})$, *and then it holds that* $\bar{R}_n(\widehat{\mathcal{G}}) \leq C \sqrt{L \sum_{\ell=1}^{L} \frac{m_{\ell+1}^{\sharp} m_{\ell}^{\sharp}}{n} \log(n)}$ *for a constant* $C > 0$.

See Appendix B.5 for the proof. Here, we again observe that there appears a trade-off between $\hat{r}$ and $m_{\ell}^{\sharp}$ because as $\tilde{r}_{\ell}$ becomes small, then $\dot{m}_{\ell}$ becomes large and thus $m_{\ell}^{\sharp}$ becomes large. The evaluation given in Theorem 3 can be substituted to the general bound (Theorem 1). If $\widehat{\mathcal{F}}$ is the full model $\mathcal{F}$, then there appears the number $\sum_{\ell=1}^{L} m_{\ell} m_{\ell+1}$ of parameters which could be larger than $n$, which is unavoidable. This dependency on the number of parameters becomes much milder if *both* of Assumptions 4 and 5 are satisfied.

**Theorem 4.** *Under Assumptions 4 and 5, it holds that*

$$\Psi(\widehat{f}) \leq \widehat{\Psi}(\widehat{f}) + C \left[ M \sqrt{\frac{[P_L \vee Q_L] L^{1 + \frac{\beta}{(2\alpha-1)} + \beta}}{n} \left( \sum_{\ell=1}^{L} m_{\ell} \right)^{\frac{4/\beta}{4/\beta + 2(1 - 1/2\alpha)}} \log(n)^3} \right.$$
$$\left. + M^{\frac{2\alpha-1}{2\alpha+1}} \left( L P_L \frac{\sum_{\ell=1}^{L} m_{\ell}}{n} \log(n) \right)^{\frac{2\alpha}{2\alpha+1}} + M \frac{R_{\mathrm{F}}^2 L^2}{R_2^2} \sqrt{\frac{\log(n)^3}{n}} + \frac{1 + Mt}{n} \right],$$

*for* $P_L = (2 L V_0 R_2^{L-1} B_x)^{1/\alpha}$ *and* $Q_L = \left[ \frac{4 U_0 R_{\mathrm{F}}^2 (1 \vee R_2)^L \exp\left(\frac{1}{4}(2\sqrt{L}-1)\right)}{(0.25)^4 (1 \wedge R_2)^{2L}} \right]^{2/\beta}$ *with probability* $1 - 3e^{-t}$ ($t > 1$), *where* $C$ *is a constant depending on* $\alpha, \beta$.

If we omit $L$ and $\log(n)$ terms for simplicity of presentation, then the bound can be written as

$$\tilde{O} \left( \sqrt{\frac{(\sum_{\ell=1}^{L} m_{\ell})^{\frac{4/\beta}{4/\beta + 2(1 - 1/2\alpha)}}}{n}} + \left( \frac{\sum_{\ell=1}^{L} m_{\ell}}{n} \right)^{\frac{2\alpha}{2\alpha+1}} \right),$$

where the $\tilde{O}(\cdot)$ symbol hides the poly-log order. This is tighter than that of Corollary 1. We can see that as $\beta$ and $\alpha$ get large, the bound becomes tighter. Actually, by taking the limit of $\alpha, \beta \to \infty$, then the bound goes to $L^2 \sqrt{\frac{\log(n)}{n}} + L \frac{\sum_{\ell=1}^{L} m_{\ell}}{n}$. Moreover, the term dependent on the width is $O(1/n)$ with respect to the sample size $n$ which is faster than the rate $O(\sqrt{\frac{\sum_{\ell=1}^{L} m_{\ell}}{n}})$ which was presented in Corollary 1. Hence, the low rank property of both the covariance matrix and the weight matrix helps to obtain better generalization. Although the bound contains the exponential term $R_2^L$, we can give a refined version that does not contain the exponential term by assuming interlayer cushion (Arora et al., 2018). See Appendix B.6 for the refined version.

There appears $\exp(\frac{1}{4}(2\sqrt{L} - 1))$ which is exponentially dependent on $L$. However, this term is moderately small for realistic settings of the depth $L$. Actually, it is 7.27 for $L = 20$ and 26.7 for $L = 50$ (we can replace this term in exchange for larger polynomial dependency on $L$). The bound is not optimized with respect to the dependency on the depth $L$. In particular, the term $L^2 \sqrt{\log(n)/n}$ could be an artifact of the proof technique and the $L^2$ term would be improved.

Finally, we compare our bound with the following norm based bounds; $O \left( \frac{R_2^L}{\sqrt{n}} \left( L \frac{R_{2 \to 1}^{2/3}}{R_2^{2/3}} \right)^{3/2} \right)$ by Bartlett et al. (2017b) and $O \left( \frac{1}{\sqrt{n}} \left( 1 + L \kappa^{\frac{4}{3}} R_{2 \to 1}^{2/3} + \kappa^{\frac{2}{3}} L R_{1 \to 1}^{2/3} \right)^{3/2} \right)$ by Wei & Ma (2019). Since our bound and their bounds are derived from different conditions, we cannot tell which is better. Here, we consider a special case where $m_{\ell} = m$ ($\forall \ell$) and $W^{(\ell)} = \frac{1}{m} \mathbf{1}\mathbf{1}^{\top} \in \mathbb{R}^{m \times m}$ which is an extreme case of low rank settings (note that $W^{(\ell)}$ has rank 1). Then, $R_2 = 1$, $R_{\mathrm{F}} = 1$,

$R_{2\to1} = \sqrt{m}$ and $R_{1\to1} = m$, and thus their bounds are $O(\sqrt{m/n})$ and $O(\sqrt{(m+m^2)/n})$ respectively. However, $\beta$ and $\alpha$ in our bound $\sqrt{m^{\frac{4/\beta}{4/\beta+2(1-1/2\alpha)}}/n} = \sqrt{m^{\frac{1}{1+\beta(1-1/2\alpha)/2}}/n}$ can be arbitrary large in this situation, so that our bound has much milder dependency on the width $m$. On the other hand, if the weight matrix has small norm and has no spectral decay (corresponding to small $\alpha$ and $\beta$), then our bound can be looser than theirs. Combining compression based bounds and norm based bounds would be interesting future work.

## 4 CONCLUSION

In this paper, we derived a compression based error bound for non-compressed network. The bound is general and it can be adopted to several compression based bound derived so far. The main difficulty lies in evaluating the population $L_2$-norm between the original network and the compressed network, but it can be overcome by utilizing the data dependent bound by the local Rademacher complexity technique. We have applied the derived bound to a situation where low rank properties of the weight matrices and the covariance matrices are assumed. The obtained bound gives much better dependency on the parameter size than ever obtained compression based ones.

**Acknowledgment** We thank the anonymous reviewers for their valuable comments. TS was partially supported by MEXT Kakenhi (15H05707, 18K19793 and 18H03201), Japan Digital Design, and JST-CREST, Japan .

## 5 PROOF OUTLINE OF THEOREM 1

Remember that for a trained network $\widehat{f}$, $\widehat{g}$ is its compressed version that is included in a submodel $\widehat{\mathcal{G}}$. First, we decompose the generalization gap as

$$\Psi(\widehat{f}) - \hat{\Psi}(\widehat{f}) = [(\Psi(\widehat{f}) - \Psi(\widehat{g})) - (\hat{\Psi}(\widehat{f}) - \hat{\Psi}(\widehat{g}))] + (\Psi(\widehat{g}) - \hat{\Psi}(\widehat{g})). \tag{4}$$

The second block in the right hand side is easy to bound, i.e., by applying the standard Rademacher complexity bound (Theorem 3.1 of Mohri et al. (2012)) with the contraction inequality (Theorem 11.6 of Boucheron et al. (2013) or Theorem 4.12 of Ledoux & Talagrand (1991)), it holds that

$$\Psi(\widehat{g}) - \hat{\Psi}(\widehat{g}) \le 2\bar{R}_n(\widehat{\mathcal{G}}) + \sqrt{M\frac{2t}{n}},$$

with probability $1 - e^{-t}$. Since $\widehat{\mathcal{G}}$ is a small model, this bound could be much smaller than a naive VC-dimension bound. The first block "bridges" the generalization gap of $\widehat{g}$ to that of $\widehat{f}$, but bounding the first block is more involved. We make use of the local Rademacher complexity to bound the term. Suppose that the event in which $\|\widehat{f} - \widehat{g}\|_{L_2} \le \dot{r}$ holds has high probability (which should be proven later), then it is also expected that $\psi(y, f) - \psi(y, g)$ has small $L_2$-norm. This is true because of the Lipschitz continuity assumption (Assumption 1). Actually, the Talagrand's concentration inequality yields that

$$\Psi(\widehat{f}) - \Psi(\widehat{g}) - (\hat{\Psi}(\widehat{f}) - \hat{\Psi}(\widehat{g})) \le C\left[\Phi(\dot{r}) + \sqrt{\dot{r}\frac{t}{n}} + \frac{1+tM}{n}\right],$$

for $\Phi(r) := \bar{R}_n(\{\psi(f) - \psi(g) \mid f \in \widehat{\mathcal{F}}, \widehat{g} \in \widehat{\mathcal{G}}, \|f - g\|_{L_2} \le \dot{r}\})$ with probability $1 - e^{-t}$. Since $\Phi(\dot{r})$ requires the restriction $\|f - g\|_{L_2} \le \dot{r}$, this quantity is much smaller than the standard Rademacher complexity $\bar{R}_n(\psi(\widehat{\mathcal{F}}) - \psi(\widehat{\mathcal{G}}))$, which yields fast convergence rate.

Finally, we should bound the probability of $\|\widehat{f} - \widehat{g}\|_{L_2} \le \dot{r}$. This can be bounded by utilizing the ratio type empirical process. Actually, we can show that

$$P\left(\sup_{h\in\widehat{\mathcal{F}}-\widehat{\mathcal{G}}} \frac{(P - P_n)(h^2)}{Ph^2 + r_*^2} \ge \frac{1}{2}\right) \le e^{-t},$$

for $r_*$ defined in Eq. (2), where $P_n f := \frac{1}{n}\sum_{i=1}^n f(z_i)$ and $Pf = \mathrm{E}[f(Z)]$. This yields that $\|\widehat{f} - \widehat{g}\|_{L_2}^2 - \|\widehat{f} - \widehat{g}\|_n^2 \le \frac{1}{2}(\|\widehat{f} - \widehat{g}\|_{L_2}^2 + r_*^2)$ with probability $1 - e^{-t}$ and equivalently $\|\widehat{f} - \widehat{g}\|_{L_2}^2 \le 2(\|\widehat{f} - \widehat{g}\|_n^2 + r_*^2)$. Since $\|\widehat{f} - \widehat{g}\|_n^2 \le \hat{r}^2$ a.s., we have $\|\widehat{f} - \widehat{g}\|_{L_2}^2 \le 2(\hat{r}^2 + r_*^2) = \dot{r}^2$.

We can show that $\Phi(\dot{r}) \le \dot{R}_{\dot{r}}(\psi(\widehat{\mathcal{F}}) - \psi(\widehat{\mathcal{G}}))$ by using the Lipschitz continuity assumption (Assumption 1). Then, we obtain the assertion.

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

## NOTATION LISTS

Since we use plenty of notations, we give the notation list in Table 2.

## APPENDIX

In the appendix, we give the proofs of the main text. We use the following notation throughout the appendix:

$$P_n f := \frac{1}{n} \sum_{i=1}^{n} f(z_i), \ \ Pf = \mathrm{E}[f(Z)].$$

To evaluate it, the *covering number* is useful (van der Vaart & Wellner, 1996).

**Definition 1** (Covering number). *For a metric space $\tilde{\mathcal{F}}$ equipped with a metric $\tilde{d}$, the $\epsilon$-covering number $\mathcal{N}(\tilde{\mathcal{F}}, \tilde{d}, \epsilon)$ is defined as the minimum number of balls with radius $\epsilon$ (measured by the metric $\tilde{d}$) to cover the metric space $\tilde{\mathcal{F}}$.*

Hereafter, $C$ denotes a constant which will be dependent on the context. We let $a \wedge b := \min\{a, b\}$ and $a \vee b := \max\{a, b\}$ for $a, b \in \mathbb{R}$.

## A  PROOF OF THEOREM 1

Denote the local Rademacher complexity of $\{\psi(f) - \psi(g) \mid f \in \widehat{\mathcal{F}}, \widehat{g} \in \widehat{\mathcal{G}}, \|f - g\|_{L_2} \leq r\}$ by

$$\Phi(r) := \bar{R}_n(\{\psi(f) - \psi(g) \mid f \in \widehat{\mathcal{F}}, \widehat{g} \in \widehat{\mathcal{G}}, \|f - g\|_{L_2} \leq r\}).$$

Here, we restate Theorem 1 in the following in more complete form. Remember that $\widehat{f} \in \widehat{\mathcal{F}}$ and $\widehat{g} \in \widehat{\mathcal{G}}$ are the trained original network and the compressed network respectively.

Table 2: Notation list

| notation | definition |
|---|---|
| $n$ | sample size |
| $z_i = (x_i, y_i)$ | $i$-th observation ($x_i$: input, $y_i$: output) |
| $D_n = (z_i)_{i=1}^n$ | training data |
| $\psi(y, f(x))$ | loss function |
| $M$ | $L_\infty$-norm bound of models |
| $B_x$ | norm bound of input |
| $\|\cdot\|_n$ | empirical $L_2$-norm ($\|f\|_n := \sqrt{\sum_{i=1}^n f(z_i)^2/n}$) |
| $\|\cdot\|_{L_2}$ | population $L_2$-norm ($\|f\|_{L_2} := \sqrt{\mathrm{E}_{Z\sim P}[f(Z)^2]}$) |
| $\widehat{\Psi}(f)$ | training error (empirical risk) |
| $\Psi(f)$ | generalization error (expected risk) |
| $(\epsilon_i)_{i=1}^n$ | Rademacher random variable |
| $\hat{R}_n(\mathcal{F}')$ | conditional Rademacher complexity |
| $\bar{R}_n(\mathcal{F}')$ | Rademacher complexity |
| $\dot{R}_r(\mathcal{F}')$ | local Rademacher complexity |
| $\hat{r}$ | upper bound of $\|\widehat{f} - \widehat{g}\|_n$ |
| $r_*$ | fixed point of the local Rademacher complexity (Eq. (2)) |
| $\dot{r}$ | $\sqrt{2(\hat{r}^2 + r_*^2)}$ |
| $L$ | depth of networks |
| $W^{(\ell)}$ | weight matrix of the $\ell$-th layer |
| $\widehat{\Sigma}_{(\ell)}$ | covariance matrix of the $\ell$-th layer |
| $R_2$ | operator norm bound of $W^{(\ell)}$ |
| $R_\mathrm{F}$ | Frobenius norm bound of $W^{(\ell)}$ |
| $\mathbf{m} = (m_1, \ldots, m_L)$ | list of widths of networks |
| $\mathbf{m}^\sharp = (m_1, m_2^\sharp, \ldots, m_L^\sharp)$ | list of widths of compressed networks |
| $\mathbf{s} = (s_1, \ldots, s_L)$ | list of ranks of the weight matrices of compressed networks |
| $\mathcal{F} = \mathrm{NN}(\mathbf{m}, R_2, R_\mathrm{F})$ | the whole set of networks with width $\mathbf{m}$ |
| $\widehat{\mathcal{F}}$ | set of trained networks |
| $\widehat{\mathcal{G}}$ | set of compressed networks |
| $\widehat{f} \in \widehat{\mathcal{F}}$ | trained network |
| $\widehat{g} \in \widehat{\mathcal{G}}$ | compressed network |
| $\dot{\mu}_j^{(\ell)}$ | an upper bound of the $j$-th largest eigenvalue of the covariance matrix in the $\ell$-th layer of $\widehat{f}$ |
| $\alpha$ | decreasing rate of the eigenvalues of $W^{(\ell)}$ |
| $\beta$ | decreasing rate of the eigenvalues of $\widehat{\Sigma}_{(\ell)}$ |

**Theorem 5.** *Suppose that the empirical $L_2$-distance between $\widehat{f}$ and $\widehat{g}$ is bounded by $\|\widehat{f} - \widehat{g}\|_n \le \hat{r}^2$ for a fixed $\hat{r} > 0$ almost surely. Let $\dot{r} := \sqrt{2(\hat{r}^2 + r_*^2)}$, then, under Assumptions 1, 2, 3, there exists a universal constant $C > 0$ such that*

$$\Psi(\widehat{f}) \le \widehat{\Psi}(\widehat{f}) + 2\bar{R}_n(\widehat{\mathcal{G}}) + \sqrt{\frac{2Mt}{n}} + C\left[\Phi(\dot{r}) + \dot{r}\sqrt{\frac{t}{n}} + \frac{1 + tM}{n}\right].$$

*with probability at least $1 - 3e^{-t}$ for all $t \ge 1$.*

Note that $\dot{R}(\psi(\widehat{\mathcal{F}}) - \psi(\widehat{\mathcal{G}}))$ in the statement of Theorem 1 of the main body is replaced by refined quantity $\Phi(\dot{r})$ (we can show $\Phi(\dot{r}) \le \dot{R}(\psi(\widehat{\mathcal{F}}) - \psi(\widehat{\mathcal{G}}))$ from the Lipschitz continuity of $\psi$).

*Proof of Theorems 1 and 5.* First, by the standard Rademacher complexity analysis, we have that

$$\Psi(\widehat{g}) - \widehat{\Psi}(\widehat{g}) = \frac{1}{n}\sum_{i=1}^n (\psi(z_i, \widehat{g}(x_i)) - \mathrm{E}[\psi(Z, \widehat{g}(X))])$$

$$\leq \sup_{g \in \widehat{\mathcal{G}}} \frac{1}{n} \sum_{i=1}^{n} (\psi(z_i, g(x_i)) - \mathrm{E}[\psi(Z, g(X))])$$

$$\leq 2\mathrm{E}_{D_n, \epsilon} \left[ \sup_{g \in \widehat{\mathcal{G}}} \frac{1}{n} \sum_{i=1}^{n} \epsilon_i \psi(z_i, g(x_i)) \right] + \sqrt{M \frac{2t}{n}} \leq 2\bar{R}_n(\widehat{\mathcal{G}}) + \sqrt{M \frac{2t}{n}} \quad (5)$$

with probability $1 - e^{-t}$, where we used the Rademacher concentration inequality (Theorem 3.1 of Mohri et al. (2012)) in the third line and the contraction inequality (Theorem 11.6 of Boucheron et al. (2013) or Theorem 4.12 of Ledoux & Talagrand (1991) and its proof) in the last line. We let this event be $\mathcal{E}_0(t)$.

Next, we observe that

$$\Psi(\widehat{f}) - \hat{\Psi}(\widehat{f}) = \Psi(\widehat{f}) - \Psi(\widehat{g}) + \Psi(\widehat{g}) - \hat{\Psi}(\widehat{g}) + \hat{\Psi}(\widehat{g}) - \hat{\Psi}(\widehat{f})$$

$$= (\Psi(\widehat{f}) - \Psi(\widehat{g}) - (\hat{\Psi}(\widehat{f}) - \hat{\Psi}(\widehat{g}))) + \Psi(\widehat{g}) - \hat{\Psi}(\widehat{g})$$

$$\leq [\Psi(\widehat{f}) - \Psi(\widehat{g}) - (\hat{\Psi}(\widehat{f}) - \hat{\Psi}(\widehat{g}))] + 2\bar{R}_n(\widehat{\mathcal{G}}) + \sqrt{M \frac{2t}{n}} \quad (6)$$

where we used Eq. (5) in the last line. Here, it should be noticed that it is not a good strategy to bound the first term $\Psi(\widehat{f}) - \Psi(\widehat{g}) - (\hat{\Psi}(\widehat{f}) - \hat{\Psi}(\widehat{g}))$ by bounding $\Psi(\widehat{f}) - \hat{\Psi}(\widehat{f})$ and $\Psi(\widehat{g}) - \hat{\Psi}(\widehat{g})$ independently. Instead, we should bound them simultaneously to obtain tighter bound. This can be accomplished by using the local Rademacher complexity technique.

Note that $\widehat{f}$ and $\widehat{g}$ are date dependent and we can only bound the empirical $L_2$-distance between them. On the other hand, the local Rademacher complexity is characterized by the population $L_2$-norm. To bridge this gap, we need to bound the population $L_2$-distance $\|\widehat{f} - \widehat{g}\|_{L_2}$ between $\widehat{f}$ and $\widehat{g}$ in terms of the empirical $L_2$-norm bound $\|\widehat{f} - \widehat{g}\|_n \leq \hat{r}$. To do so, we also bound the local Rademacher complexity of $\widehat{\mathcal{F}} - \widehat{\mathcal{G}}$: $\dot{R}_r(\widehat{\mathcal{F}} - \widehat{\mathcal{G}})$. Suppose that there exists a function $\phi : [0, \infty) \to [0, \infty)$ such that the the following conditions are satisfied:

$$\dot{R}_r(\widehat{\mathcal{F}} - \widehat{\mathcal{G}}) \leq \phi(r)$$

and

$$\phi(2r) \leq 2\phi(r).$$

Note that Eq. (11) gives one example of $\phi(r)$. Then, by the so-called *peeling device*, we can show that for any $r > 0$,

$$P \left( \sup_{h \in \widehat{\mathcal{F}} - \widehat{\mathcal{G}}} \frac{(P - P_n)(h^2)}{Ph^2 + r^2} \geq 8 \frac{\phi(r)}{r^2} + M\sqrt{\frac{4t}{r^2 n}} + M^2 \frac{2t}{r^2 n} \right) \leq e^{-t}$$

for all $t > 0$ (Theorem 7.7 and Eq. (7.17) of Steinwart & Christmann (2008)). Hence, if we choose $r_* = r_*(t)$ so that

$$8 \frac{\phi(r_*)}{r_*^2} + M\sqrt{\frac{4t}{r_*^2 n}} + M^2 \frac{2t}{r_*^2 n} \leq \frac{1}{2},$$

then it holds that

$$P(h^2) \leq 2P_n(h^2) + 2r_*^2$$

uniformly over all $h \in \widehat{\mathcal{F}} - \widehat{\mathcal{G}}$ with probability greater than $1 - e^{-t}$. We let this event as $\mathcal{E}_1(t)$. In this event, if $\|\widehat{f} - \widehat{g}\|_n^2 \leq \hat{r}^2$, then it holds that

$$\|\widehat{f} - \widehat{g}\|_{L_2}^2 \leq 2(\hat{r}^2 + r_*^2) = \dot{r}^2.$$

Next, we bound $\Psi(\widehat{f}) - \Psi(\widehat{g}) - (\hat{\Psi}(\widehat{f}) - \hat{\Psi}(\widehat{g}))$. To bound this term, we apply the Talagrand's concentration inequality (Proposition 2 and Talagrand (1996); Bousquet (2002)). To apply it, we should bound the variance and $L_\infty$-norm of $\psi(y, f(x)) - \psi(y, g(x)) - (\mathrm{E}[\psi(Y, f(X))] - \mathrm{E}[\psi(Y, g(X))])$ for any $f \in \widehat{\mathcal{F}}, g \in \widehat{\mathcal{G}}$ with $\|f - g\|_{L_2} \leq r$ (where $r$ will be set $2(\hat{r}^2 + r_*^2)$). Due to the Lipschitz continuity of $\psi$, we have that

$$\mathrm{Var}[\psi(Y, f(X)) - \psi(Y, g(X))] \leq \mathrm{Var}[f(X) - g(X)] \leq \|f - g\|_{L_2}^2 \leq r^2.$$

Similarly, it holds that

$$
\begin{aligned}
&|\psi(y, f(x)) - \psi(y, g(x)) - (\mathrm{E}[\psi(Y, f(X))] - \mathrm{E}[\psi(Y, g(X))])| \\
&\leq |\psi(y, f(x)) - \mathrm{E}[\psi(Y, f(X))]| + |\psi(y, g(x)) - \mathrm{E}[\psi(Y, g(X))]| \leq 2M.
\end{aligned}
$$

Hence, by the Talagrand's concentration inequalit (Proposition 2 and Talagrand (1996); Bousquet (2002)), it holds that

$$
\begin{aligned}
&\sup_{f,g:\|f-g\|_{L_2}\leq r} (P - P_n)(\psi(Y, f(X)) - \psi(Y, g(X))) \\
&\leq 2\mathrm{E}\left[\sup_{f,g:\|f-g\|_{L_2}\leq r} (P - P_n)(\psi(Y, f(X)) - \psi(Y, g(X)))\right] + r\sqrt{\frac{2t}{n}} + \frac{4tM}{n},
\end{aligned}
$$

with probability at least $1 - e^{-t}$ for any $t > 0$. The first term in the right hand side can be bounded as

$$
\begin{aligned}
&\mathrm{E}\left[\sup_{f,g:\|f-g\|_{L_2}\leq r} (P - P_n)(\psi(Y, f(X)) - \psi(Y, g(X)))\right] \\
&\leq 2\mathrm{E}_{D_n,\epsilon}\left[\sup_{f,g:\|f-g\|_{L_2}\leq r} \frac{1}{n}\sum_{i=1}^{n} \epsilon_i[\psi(y_i, f(x_i)) - \psi(y_i, g(x_i))]\right] \\
&= 2\Phi(n),
\end{aligned}
$$

where we used the standard symmetrization argument (see Lemma 11.4 of Boucheron et al. (2013) for example).

Combining these inequalities, it holds that

$$
\sup_{f,g:\|f-g\|_{L_2}\leq r} (P - P_n)(\psi(Y, f(X)) - \psi(Y, g(X))) \leq C\left[\Phi(r) + r\sqrt{\frac{t}{n}} + \frac{1 + Mt}{n}\right].
$$

for a universal constant $C > 0$ with probability at least $1 - e^{-t}$ for all $t > 0$. We denote by this event as $\mathcal{E}_2(t, r)$.

We define an event $\mathcal{E}_3(t) = \mathcal{E}_1(t) \cap \mathcal{E}_2(t, \dot{r})$. Then $P(\mathcal{E}_3(t)) \geq 1 - 2e^{-t}$ for all $t > 0$. In this event, $\|\widehat{f} - \widehat{g}\|_{L_2}^2 \leq \dot{r}^2$ and thus it holds that

$$
\Psi(\widehat{f}) - \Psi(\widehat{g}) - (\hat{\Psi}(\widehat{f}) - \hat{\Psi}(\widehat{g})) \leq C\left[\Phi(\dot{r}) + \dot{r}\sqrt{\frac{t}{n}} + \frac{1 + tM}{n}\right],
$$

for a universal constant $C > 0$. Combining this and Eq. (6), we obtain the assertion on the event $\mathcal{E}_0(t) \cap \mathcal{E}_3(t)$. This gives the proof of Theorem 5.

To show Theorem 1, note that $\|\psi(f) - \psi(g)\|_{L_2} \leq \|f - g\|_{L_2}$ by the Lipschitz continuity of $\psi$ and this yields $\{\psi(f) - \psi(g) \mid f \in \widehat{\mathcal{F}}, g \in \widehat{\mathcal{G}}, \|f - g\|_{L_2} \leq r\} \subset \{\psi(f) - \psi(g) \mid f \in \widehat{\mathcal{F}}, g \in \widehat{\mathcal{G}}, \|\psi(f) - \psi(g)\|_{L_2} \leq r\}$. Therefore, we have

$$
\Phi(\dot{r}) \leq \dot{R}_{\dot{r}}(\psi(\widehat{\mathcal{F}}) - \psi(\widehat{\mathcal{G}})).
$$

□

Hereafter, we derive some upper bounds of the (local) Rademacher complexities under some covering number conditions.

**Lemma 1.** *For a given $r > 0$, let $\hat{\gamma}_n = \hat{\gamma}_n(D_n) := \sup\{\|f - g\|_n \mid \|f - g\|_{L_2} \leq r, f \in \widehat{\mathcal{F}}, g \in \widehat{\mathcal{G}}\}$. Then, it holds that*

$$
\begin{aligned}
&\max\{\Phi(r), \dot{R}_r(\widehat{\mathcal{F}} - \widehat{\mathcal{G}})\} \\
&\leq C\left\{\frac{1}{n} + \mathrm{E}_{D_n}\left[\int_{1/n}^{\hat{\gamma}_n} \sqrt{\frac{\log(\mathcal{N}(\widehat{\mathcal{F}}, \|\cdot\|_n, \epsilon/2))}{n}} d\epsilon + \int_{1/n}^{\hat{\gamma}_n} \sqrt{\frac{\log(\mathcal{N}(\widehat{\mathcal{G}}, \|\cdot\|_n, \epsilon/2))}{n}} d\epsilon\right]\right\},
\end{aligned}
$$
(7)

*and*

$$\Phi(r) \leq C' \dot{R}_r(\psi(\widehat{\mathcal{F}}) - \psi(\widehat{\mathcal{G}}))\sqrt{\log(n)}\log(2nM), \tag{8}$$

*where $C, C'$ are universal constants.*

*Proof.* The conditional Rademacher complexity of the set $\{\psi(y, f(x)) - \psi(y, g(x)) \mid f \in \widehat{\mathcal{F}}, g \in \widehat{\mathcal{G}}, \|f - g\|_{L_2} \leq r\}$ can be bounded by a constant times the following *Dudley integral* (see Theorem 5.22 of Wainwright (2019) or Lemma A.5 of Bartlett et al. (2017a) for example):

$$\inf_{\alpha > 0}\left[\alpha + \int_\alpha^{\hat{\gamma}_n} \sqrt{\frac{\log(\mathcal{N}(\{\psi(f) - \psi(g) \mid f \in \widehat{\mathcal{F}}, g \in \widehat{\mathcal{G}}, \|f - g\|_{L_2} \leq r\}, \|\cdot\|_n, \epsilon))}{n}}d\epsilon\right]$$

$$\leq \frac{1}{n} + \int_{1/n}^{\hat{\gamma}_n} \sqrt{\frac{\log(\mathcal{N}(\{\psi(f) - \psi(g) \mid f \in \widehat{\mathcal{F}}, g \in \widehat{\mathcal{G}}, \|f - g\|_{L_2} \leq r\}, \|\cdot\|_n, \epsilon))}{n}}d\epsilon \tag{9}$$

$$\leq \frac{1}{n} + \int_{1/n}^{\hat{\gamma}_n} \sqrt{\frac{\log(\mathcal{N}(\psi(\widehat{\mathcal{F}}), \|\cdot\|_n, \epsilon/2)) + \log(\mathcal{N}(\psi(\widehat{\mathcal{G}}), \|\cdot\|_n, \epsilon/2))}{n}}d\epsilon$$

$$\leq \frac{1}{n} + \int_{1/n}^{\hat{\gamma}_n} \sqrt{\frac{\log(\mathcal{N}(\widehat{\mathcal{F}}, \|\cdot\|_n, \epsilon/2)) + \log(\mathcal{N}(\widehat{\mathcal{G}}, \|\cdot\|_n, \epsilon/2))}{n}}d\epsilon$$

$$\leq \frac{1}{n} + \int_{1/n}^{\hat{\gamma}_n} \sqrt{\frac{\log(\mathcal{N}(\widehat{\mathcal{F}}, \|\cdot\|_n, \epsilon/2))}{n}}d\epsilon + \int_{1/n}^{\hat{\gamma}_n} \sqrt{\frac{\log(\mathcal{N}(\widehat{\mathcal{G}}, \|\cdot\|_n, \epsilon/2))}{n}}d\epsilon, \tag{10}$$

where we used $\|\psi(f) - \psi(g) - (\psi(f') - \psi(g'))\|_n \leq \|\psi(f) - \psi(f')\|_n + \|\psi(g) - \psi(g')\|_n \leq \epsilon$ for $f, f' \in \widehat{\mathcal{F}}$ and $g, g' \in \widehat{\mathcal{G}}$ with $\|f - f'\|_n \leq \epsilon/2$ and $\|g - g'\|_n \leq \epsilon/2$ in the third line, and 1-Lipschitz continuity of the loss function $\psi$ in the fourth line (i.e., $|\psi(y, f(x)) - \psi(y, g(x))| \leq |f(x) - g(x)|$ which yields $\|\psi(f) - \psi(g)\|_n \leq \|f - g\|_n$),

In the same way, we can see that $\dot{R}_r(\widehat{\mathcal{F}} - \widehat{\mathcal{G}})$ is bounded by the Dudley integral as

$$\dot{R}_r(\widehat{\mathcal{F}} - \widehat{\mathcal{G}})$$

$$\leq \frac{C}{n} + C\mathbb{E}_{D_n}\left[\int_{1/n}^{\hat{\gamma}_n} \sqrt{\frac{\log(\mathcal{N}(\{\hat{f} - \hat{g} \mid \hat{f} \in \widehat{\mathcal{F}}, \hat{g} \in \widehat{\mathcal{G}}, \|\hat{f} - \hat{g}\|_{L_2} \leq r\}, \|\cdot\|_n, \epsilon))}{n}}d\epsilon\right]$$

$$\leq \frac{C}{n} + C\mathbb{E}_{D_n}\left[\int_{1/n}^{\hat{\gamma}_n} \sqrt{\frac{\log(\mathcal{N}(\widehat{\mathcal{F}}, \|\cdot\|_n, \epsilon/2))}{n}}d\epsilon + \int_{1/n}^{\hat{\gamma}_n} \sqrt{\frac{\log(\mathcal{N}(\widehat{\mathcal{G}}, \|\cdot\|_n, \epsilon/2))}{n}}d\epsilon\right], \tag{11}$$

where we used the same argument as Eq. (10) and $C > 0$ is a universal constant. Then, we conclude Eq. (7).

Next, we show Eq. (8). The term Eq. (9) can be evaluated by using the local Rademacher complexity of $\psi(\widehat{\mathcal{F}}) - \psi(\widehat{\mathcal{G}})$. Note that $\|\psi(f) - \psi(g)\|_{L_2} \leq \|f - g\|_{L_2}$ by the Lipschitz continuity of $\psi$ and this yields $\{\psi(f) - \psi(g) \mid f \in \widehat{\mathcal{F}}, g \in \widehat{\mathcal{G}}, \|f - g\|_{L_2} \leq r\} \subset \{\psi(f) - \psi(g) \mid f \in \widehat{\mathcal{F}}, g \in \widehat{\mathcal{G}}, \|\psi(f) - \psi(g)\|_{L_2} \leq r\}$. Then, the Sudakov's minoration (Corollary 4.14 of Ledoux & Talagrand (1991)) gives an upper bound of the right hand side of Eq. (9):

$$\int_{1/n}^{\hat{\gamma}_n} \sqrt{\frac{\log(\mathcal{N}(\{\psi(f) - \psi(g) \mid f \in \widehat{\mathcal{F}}, g \in \widehat{\mathcal{G}}, \|f - g\|_{L_2} \leq r\}, \|\cdot\|_n, \epsilon))}{n}}d\epsilon$$

$$\leq \int_{1/n}^{\hat{\gamma}_n} \sqrt{\frac{\log(\mathcal{N}(\{\psi(f) - \psi(g) \mid f \in \widehat{\mathcal{F}}, g \in \widehat{\mathcal{G}}, \|\psi(f) - \psi(g)\|_{L_2} \leq r\}, \|\cdot\|_n, \epsilon))}{n}}d\epsilon$$

$$\leq \int_{1/n}^{\hat{\gamma}_n} \hat{R}_{n,r}(\psi(\widehat{\mathcal{F}}) - \psi(\widehat{\mathcal{G}}))\sqrt{\log(n)}\frac{1}{\epsilon}d\epsilon \leq \hat{R}_{n,r}(\psi(\widehat{\mathcal{F}}) - \psi(\widehat{\mathcal{G}}))\sqrt{\log(n)}\log(n\hat{\gamma}_n),$$

where $\hat{R}_{n,r}(\psi(\widehat{\mathcal{F}})-\psi(\widehat{\mathcal{G}})) := \mathrm{E}_\epsilon\left[\sup\{\frac{1}{n}\sum_{i=1}^n \epsilon_i(\psi(y_i,f(x_i))-\psi(y_i,g(x_i))) \mid f \in \widehat{\mathcal{F}}, g \in \widehat{\mathcal{G}}, \|\psi(f)-\psi(g)\|_{L_2} \le r\}\right]$.

Since $\hat{\gamma}_n \le 2M$, the expectation of the right hand side with respect to $D_n$ is $\dot{R}_r(\psi(\widehat{\mathcal{F}}) - \psi(\widehat{\mathcal{G}}))\sqrt{\log(n)}\log(2nM)$. This gives an upper bound of the right hand side of Eq. (9) and yields Eq. (8). $\qquad\square$

**Lemma 2.**
$$\mathrm{E}\left[\sup\{P_n h^2 \mid h \in \widehat{\mathcal{F}} - \widehat{\mathcal{G}} : \|h\|_{L_2} \le r\}\right] \le r^2 + 2M\phi(r).$$

*Proof.* By the contraction inequality of the Rademacher complexity (Theorem 4.12 of Ledoux & Talagrand (1991) and its proof), we have

$$\mathrm{E}\left[\sup\{P_n h^2 \mid h \in \widehat{\mathcal{F}} - \widehat{\mathcal{G}} : \|h\|_{L_2} \le r\}\right]$$

$$\le \mathrm{E}\left[\sup\{(P_n - P)h^2 \mid h \in \widehat{\mathcal{F}} - \widehat{\mathcal{G}} : \|h\|_{L_2} \le r\}\right] + r^2$$

$$\le 2\mathrm{E}\left[\sup\{\frac{1}{n}\sum_{i=1}^n \epsilon_i h(x_i)^2 \mid h \in \widehat{\mathcal{F}} - \widehat{\mathcal{G}} : \|h\|_{L_2} \le r\}\right] + r^2$$

(symmetrization; Lemma 11.4 of Boucheron et al. (2013))

$$\le 2M\mathrm{E}\left[\sup\{\frac{1}{n}\sum_{i=1}^n \epsilon_i h(x_i) \mid h \in \widehat{\mathcal{F}} - \widehat{\mathcal{G}} : \|h\|_{L_2} \le r\}\right] + r^2 \quad (\because \text{contraction inequality})$$

$$= 2M\dot{R}_r(\widehat{\mathcal{F}} - \widehat{\mathcal{G}}) + r^2 \le 2M\phi(r) + r^2.$$

$\qquad\square$

**Lemma 3.** *Suppose that*
$$\sup_{D_n} \log(\mathcal{N}(\widehat{\mathcal{F}}, \|\cdot\|_n, \epsilon/2)) + \sup_{D_n} \log(\mathcal{N}(\widehat{\mathcal{G}}, \|\cdot\|_n, \epsilon/2)) \le S_1 + S_2\log(1/\epsilon) + S_3\epsilon^{-2q}$$

*for $q < 1$. Then, for a universal constant $C > 0$ and a constant $C_q > 0$ which depends on $q < 1$, it holds that*

$$\Phi(r) \le C\max\left\{\frac{1}{n} + M\frac{S_1 + S_2\log(n)}{n} + r\sqrt{\frac{S_1 + S_2\log(n)}{n}},\right.$$
$$\left. C_q\left[\frac{1}{n} + \left(\frac{M^{1-q}S_3}{n}\right)^{\frac{1}{1+q}} + r^{1-q}\sqrt{\frac{S_3}{n}}\right]\right\}.$$

*In particular,*
$$r_*^2 \le C\left[M\frac{S_1 + S_2\log(n)}{n} + \left(\frac{M^{1-q}S_3}{n}\right)^{\frac{1}{1+q}} + \frac{1 + Mt}{n}\right].$$

*Proof.* Remember that $\hat{\gamma}_n := \sup\{\|f - g\|_n : \|f - g\|_{L_2} \le r, f \in \widehat{\mathcal{F}}, g \in \widehat{\mathcal{G}}\}$ for a given $r > 0$. Under the assumption, we may take $\phi(r)$ (defined below) as an upper bound of $\Phi(r)$ by Lemma 1:

$$\phi(r) = C\left(\frac{1}{n} + \frac{1}{\sqrt{n}}\mathrm{E}\left[\int_{1/n}^{\hat{\gamma}_n}\sqrt{S_1 + S_2\log(\epsilon^{-1}) + S_3\epsilon^{-2q}}\,\mathrm{d}\epsilon\right]\right),$$

where $C > 0$ is a universal constant. The right hand side can be evaluated as

$$\mathrm{E}\left[\int_{1/n}^{\hat{\gamma}_n}\sqrt{S_1 + S_2\log(1/\epsilon) + S_3\epsilon^{-2q}}\,\mathrm{d}\epsilon\right] \le \mathrm{E}\left[\hat{\gamma}_n\sqrt{S_1 + S_2\log(n)}\right] + \frac{1}{1-q}\sqrt{S_3}\mathrm{E}[\hat{\gamma}_n^{1-q}]$$

$$\le \sqrt{\mathrm{E}[\{P_n h^2 \mid h \in \widehat{\mathcal{F}} - \widehat{\mathcal{G}}, \|h\|_{L_2} \le r\}]}\sqrt{S_1 + S_2\log(n)}$$

$$+ \frac{\sqrt{S_3}}{1-q}\mathrm{E}[\{P_n h^2 \mid h \in \widehat{\mathcal{F}} - \widehat{\mathcal{G}}, \|h\|_{L_2} \le r\}]^{\frac{1-q}{2}}$$

$$\leq \sqrt{r^2 + 2M\phi(r)}\sqrt{S_1 + S_2\log(n)} + \frac{\sqrt{S_3}}{1-q}(r^2 + 2M\phi(r))^{\frac{1-q}{2}}, \tag{12}$$

where we used Lemma 2. Hence, if the first term is larger than the second term, we have that

$$\phi(r) \leq C\left(\frac{1}{n} + \sqrt{\frac{S_1 + S_2\log(n)}{n}}\sqrt{r^2 + 2M\phi(r)}\right)$$

$$\leq \frac{C}{n} + C^2 M\frac{S_1 + S_2\log(n)}{n} + Cr\sqrt{\frac{S_1 + S_2\log(n)}{n}} + \frac{\phi(r)}{2}.$$

Therefore, we obtain that

$$\phi(r) \leq \frac{2C}{n} + 2C^2 M\frac{S_1 + S_2\log(n)}{n} + 2Cr\sqrt{\frac{S_1 + S_2\log(n)}{n}}. \tag{13}$$

On the other hand, if the second term in Eq. (12) is larger than the first one, then Young's inequality gives that

$$\phi(r) \leq C\left(\frac{1}{n} + \sqrt{\frac{S_3}{n(1-q)^2}}(r^2 + 2M\phi(r))^{\frac{1-q}{2}}\right)$$

$$\leq \frac{C}{n} + C\left[q\left(\frac{c'^{1-q}C^2 S_3}{n(1-q)^2}\right)^{\frac{1}{1+q}} + (1-q)\frac{2M\phi(r)}{c'} + \sqrt{\frac{S_3}{n(1-q)^2}r^{2(1-q)}}\right].$$

where $c' > 0$ is any positive real. Thus taking $c'$ sufficiently large (which depends on $M, q$), we conclude that

$$\phi(r) \leq C_q\left[\frac{1}{n} + \left(\frac{M^{1-q}S_3}{n}\right)^{\frac{1}{1+q}} + \sqrt{\frac{S_3}{n}r^{2(1-q)}}\right], \tag{14}$$

where $C_q$ is a constant depending only on $q < 1$. These two inequalities (Eq. (13) and Eq. (14)) give the first assertion. By noticing the assumption $M \geq 1$, $r_*$ can be derived from a simple calculation.

$\square$

# B DERIVATION OF COMPRESSION BASED BOUND FOR NON-COMPRESSED NETWORKS

## B.1 FULL MODEL BOUND

Here, we assume that the model $\widehat{\mathcal{F}}$ of the trained network is the full model $\widehat{\mathcal{F}} = \mathcal{F} = \mathrm{NN}(\mathbf{m}, R_2, R_F)$ and $\widehat{\mathcal{G}}$ is included in $\mathrm{NN}(\mathbf{m}, \hat{R}_2, \hat{R}_F)$. Then, their covering entropy is bounded by

$$\log(\mathcal{N}(\widehat{\mathcal{F}}, \|\cdot\|_\infty, \epsilon)) \leq (\sum_{\ell=1}^{L} m_\ell m_{\ell+1})\log(\epsilon^{-1}) + L(\sum_{\ell=1}^{L} m_\ell m_{\ell+1})\log(L(R_2 \vee 1)(\max_\ell m_\ell + 1)),$$

$$\log(\mathcal{N}(\widehat{\mathcal{G}}, \|\cdot\|_\infty, \epsilon)) \leq (\sum_{\ell=1}^{L} m_\ell m_{\ell+1})\log(\epsilon^{-1}) + L(\sum_{\ell=1}^{L} m_\ell m_{\ell+1})\log(L(\hat{R}_2 \vee 1)(\max_\ell m_\ell + 1)).$$

Hence, the condition in Lemma 3 holds for $S_1 = \sum_{\ell=1}^{L} m_\ell m_{\ell+1}$, $S_2 = LS_1\log(L(R_2 \vee \hat{R}_2 \vee 1)(\max_\ell m_\ell + 1))$ and $S_3 = 0$. In this case, we can set

$$r_*^2 = C\frac{(M+1)(S_1 + 1 + S_2\log(n)) + Mt}{n}$$

for a constant $C > 0$.

## B.2 COMPLEXITY OF A SPARSE MODEL (PROOF OF EXAMPLE 1)

Suppose that $\widehat{\mathcal{G}}$ is the model with sparse weight matrices given in Example 1. Let $m = \max_\ell m_\ell$ and $B = R_2$, then we can see that

$$\widehat{\mathcal{G}} \subset \Phi(L, m, S, B)$$

where the definition of $\Phi(L, m, S, B)$ is given in Appendix C.1. Therefore, its covering number is bounded by

$$\log(\mathcal{N}(\widehat{\mathcal{G}}, \|\cdot\|_\infty, \epsilon))$$
$$\leq S \log(\epsilon^{-1}) + LS \log(L(R_2 \vee 1)(\max_\ell m_\ell + 1)) \leq O(SL \log(n) + S \log(\epsilon^{-1})),$$

by Lemma 4. Hence, the Rademacher complexity is bounded as

$$\widehat{\mathcal{G}} \leq CM \sqrt{L\frac{S}{n} \log(nL(R_2 \vee 1)(\max_\ell m_\ell + 1))} = O\left(M\sqrt{L\frac{S}{n}\log(n)}\right),$$

by the Dudley integral, $\bar{R}(\widehat{\mathcal{G}}) \lesssim \int_0^M \sqrt{\frac{\log(\mathcal{N}(\widehat{\mathcal{G}}, \|\cdot\|_\infty, \epsilon))}{n}} \mathrm{d}\epsilon$, where $C$ is a universal constant.

## B.3 NEAR LOW RANK CONDITION ON THE WEIGHT MATRIX (PROOF OF THEOREM 2 AND COROLLARY 1 )

Here, we give proofs of Theorem 2 and Corollary 1 which give a generalization error bound when the trained network has near low rank weight matrices $(W^{(\ell)})_{\ell=1}^L$ (Assumption 4).

Under Assumption 4, we can see that for any $1 \leq s \leq \min\{m_\ell, m_{\ell+1}\}$, we can approximate $W^{(\ell)}$ by a rank $s$ matrix $W'$ as

$$\|W^{(\ell)} - W'\|_2 \leq V_0 s^{-\alpha}, \quad \|W'\|_2 \leq \|W^{(\ell)}\|_2 \tag{15}$$

$$\|W^{(\ell)} - W'\|_F \leq \frac{1}{\sqrt{2\alpha - 1}} V_0 (s-1)^{(1-\alpha)/2}. \tag{16}$$

This can be checked by discarding the singular vectors corresponding to the singular values smaller than the $s$-th largest one. This ensures that, for any $f \in \widehat{\mathcal{F}}$, there exists $f' \in \mathcal{F}$ such that it has width $\mathbf{s} = (s_1, \ldots, s_L)$, weight matrix $W^{\sharp(\ell)}$ with $\|W^{\sharp(\ell)}\|_2 \leq R_2$, $\|W^{\sharp(\ell)}\|_F \leq R_F$ and

$$\|f - f'\|_\infty \leq \sum_{\ell=1}^L V_0 R_2^{L-1} s_\ell^{-\alpha} B_x. \tag{17}$$

This can be proved as follows. Let $f'(x) = G \circ (W^{\sharp(L)} \eta(\cdot)) \circ \cdots \circ (W^{\sharp(1)} x)$ where $W^{\sharp(\ell)}$ is a rank $s_\ell$ matrix that satisfies Eqs. (15) and (16) for $W' = W^{\sharp(\ell)}$. Let $f_\ell(x) = G \circ (W^{\sharp(L)} \eta(\cdot)) \circ \cdots \circ (W^{\sharp(\ell+1)} \eta(\cdot)) \circ (W^{(\ell)} \eta(\cdot)) \circ \cdots \circ (W^{(1)} x)$ and $f_0(x) = f'$. Then, $\|f - f'\|_\infty \leq \sum_{\ell=1}^L \|f_\ell - f_{\ell-1}\|_\infty$. We can see that $\|(W^{(\ell)} \eta(\cdot)) \circ \cdots \circ (W^{(1)} x)\| \leq \prod_{k=1}^\ell \|W^{(k)}\|_2 \|x\| \leq R_2^\ell B_x$, $\|(W^{\sharp(\ell)} \eta(\cdot)) \circ (W^{(\ell-1)} \eta(\cdot)) \circ \cdots \circ (W^{(1)} x) - (W^{(\ell)} \eta(\cdot)) \circ (W^{(\ell-1)} \eta(\cdot)) \circ \cdots \circ (W^{(1)} x)\| \leq \|W^{(\ell)} - W^{\sharp(\ell)}\|_2 \|(W^{(\ell-1)} \eta(\cdot)) \circ \cdots \circ (W^{(1)} x)\| \leq V_0 s_\ell^{-\alpha} R_2^{\ell-1} B_x$. This gives $\|f_\ell - f_\ell\|_\infty \leq \prod_{k=\ell+1}^L \|W^{(k)}\|_2 V_0 s_\ell^{-\alpha} R_2^{\ell-1} B_x \leq R_2^{L-1} V_0 s_\ell^{-\alpha} B_x$. Finally, by summing up this from $\ell = 1$ to $\ell = L$, we obtain Eq. (17).

In particular, for any $\epsilon > 0$, by setting $s_\ell' = s_\ell'(\epsilon) = \min\{\lceil (\epsilon/(LV_0 R_2^{L-1} B_x))^{-1/\alpha} \rceil, m_\ell \wedge m_{\ell+1}\}$ for all $\ell$, then $\|f - f'\|_\infty \leq \epsilon$. This indicates that, by Lemma 5, the covering entropy of $\widehat{\mathcal{F}}$ is bounded by

$$\log(\mathcal{N}(\widehat{\mathcal{F}}, \|\cdot\|_\infty, \epsilon)) \leq \left(\sum_{\ell=1}^L (m_\ell + m_{\ell+1}) s_\ell'(\epsilon/2)\right) [\log(\epsilon^{-1}) + 2L \log(2L(R_2 \vee 1)(\max_\ell m_\ell + 1))] \tag{18}$$

$$\leq 2(\sum_{\ell=1}^L m_\ell)(2LV_0 R_2^{L-1} B_x)^{1/\alpha} \epsilon^{-1/\alpha} [\log(\epsilon^{-1}) + 2L \log(2L(R_2 \vee 1)(\max_\ell m_\ell + 1))].$$

As the compressed network $\widehat{\mathcal{G}}$, we may choose $\widehat{\mathcal{G}} = \mathrm{NN}(\mathbf{m}, \mathbf{s}, R_2, R_{\mathrm{F}})$ for $\mathbf{s} = (s_1, \ldots, s_L)$ so that, for all $f \in \widehat{\mathcal{F}}$, there exists $g \in \widehat{\mathcal{G}}$ satisfying

$$\|f - g\|_\infty \le (V_0 R_2^{L-1} B_x) \sum_{\ell=1}^{L} s_\ell^{-\alpha}.$$

Hence, we may set $\hat{r}^2 = [(V_0 R_2^{L-1} B_x) \sum_{\ell=1}^{L} s_\ell^{-\alpha}]^2$. In this case, the covering number of $\widehat{\mathcal{G}}$ is bounded as (18) by replacing $s'_\ell$ with $s_\ell$.

Therefore, Lemma 3 gives that

$$r_*^2 \le C \frac{(M+1)(S_1 + 1 + S_2 \log(n)) + Mt}{n} \vee M^{\frac{2\alpha-1}{2\alpha+1}} \left(\frac{S_3}{n}\right)^{\frac{2\alpha}{1+2\alpha}},$$

where

$$S_1 = \sum_{\ell=1}^{L} s_\ell(m_\ell + m_{\ell+1}),$$

$$S_2 = LS_1 \log(L(R_2 \vee 1)(\max_\ell m_\ell + 1)),$$

$$S_3 = (\sum_{\ell=1}^{L} m_\ell)(2LV_0 R_2^{L-1} B_x)^{1/\alpha}[\log(n) + 2L\log(2L(R_2 \vee 1)(\max_\ell m_\ell + 1))],$$

where $q = 1/2\alpha$ was used. This indicates that, if $\alpha > 1/2$ is large (in other words, each weight matrix is close to rank 1), then the local Rademacher complexity can be small. Actually, the bound is smaller than $\frac{\sum_{\ell=1}^{L} m_\ell m_{\ell+1}}{n}$ because each rank $s_\ell$ must satisfy $s_\ell \le \min\{m_\ell, m_{\ell+1}\}$.

Finally, we observe that

$$\bar{R}_n(\widehat{\mathcal{G}}) \le CM\sqrt{L\frac{\sum_{\ell=1}^{L} s_\ell(m_\ell + m_{\ell+1})}{n} \log(nL(R_2 \vee 1)(\max_\ell m_\ell + 1))},$$

by Lemma 5 for $\widehat{\mathcal{G}} = \mathrm{NN}(\mathbf{m}, \mathbf{s}, R_2, R_{\mathrm{F}})$ and the Dudley integral (van der Vaart & Wellner, 1996): $\bar{R}(\widehat{\mathcal{G}}) \lesssim \int_0^M \sqrt{\frac{\log(\mathcal{N}(\widehat{\mathcal{G}}, \|\cdot\|_\infty, \epsilon))}{n}} d\epsilon$. This gives Theorem 2.

Corollary 1 can be obtained by substituting $s_\ell = \min\{m_\ell, m_{\ell+1}, \lceil LV_0 R_2^{L-1} B_x \rceil^{1/\alpha}\}$.

### B.4 IMPROVED BOUND WITH LIPSCHITZ CONTINUITY CONSTRAINT

In the generalization error bound of Theorem 2 and Corollary 1, there appears $R_2^L$. Even though $R_2$ can be much smaller than $R_{\mathrm{F}}$, the exponential dependency $R_2^L$ could give lose bound as pointed out in Arora et al. (2018). We improve this exponential dependency by assuming the following condition.

**Assumption 6** (Lipschitz continuity between layers: Interlayer cushion, interlayer smoothness (Arora et al., 2018)). *For the trained network $\widehat{f} = G \circ (W^{(L)}\eta(\cdot)) \circ \cdots \circ (W^{(1)}x)$, let $\phi_\ell(x) = \eta \circ (W^{(\ell-1)}\eta(\cdot)) \circ \cdots \circ (W^{(1)}x)$ be the input to the $\ell$-th layer and $M_{\ell,\ell'}(x) = (W^{(\ell')}\eta(\cdot)) \circ \cdots \circ (W^{(\ell)}x)$ be the transformation from the $\ell$-th layer to $\ell'$-th layer. Then, we assume that there exists $\kappa, \tau > 0$ such that $\tau \le 1/(2\kappa^2 L)$ and for any $\ell, \ell' \in [L]$,*

$$\sum_{i=1}^{n}[M_{\ell,\ell'}(\phi_\ell(x_i) + \xi_i^{(1)}) - M_{\ell,\ell'}(\phi_\ell(x_i) + \xi_i^{(1)} + \xi_i^{(2)})]^2 \le \kappa^2 \left(\|\xi^{(1)}\| + \tau\|\xi^{(2)}\|\right)^2,$$

*for all $\xi^{(1)} = (\xi_1^{(1)}, \ldots, \xi_n^{(1)})^\top \in \mathbb{R}^n$ and $\xi^{(2)} = (\xi_1^{(2)}, \ldots, \xi_n^{(2)})^\top \in \mathbb{R}^n$.*

This assumption is a simplified version of the interlayer cushion and the interlayer smoothness introduced in Arora et al. (2018). Although a trivial bound of $\kappa$ is $\kappa \leq R_2^L$, the practically observed Lipschitz constant is usually much smaller. Assumption 6 captures this point and gives better dependency on the depth $L$. Actually, we can remove the exponential dependency on $R_2$ as in the following corollary.

**Corollary 2.** *Under Assumptions 4 and 6, it holds that*

$$\Psi(\widehat{f}) \leq \hat{\Psi}(\widehat{f}) + C\left[M^{1-1/2\alpha}\sqrt{L\frac{(\sum_{\ell=1}^L m_\ell)(2LV_0\kappa^2 B_x)^{1/\alpha}}{n}\log(n)} + M^{\frac{2\alpha-1}{2\alpha+1}}A_2'^{\frac{2\alpha}{2\alpha+1}} + \frac{1+tM}{n}\right]$$

*for $A_2' = L\frac{(\sum_{\ell=1}^L m_\ell)(2LV_0\kappa^2 B_x)^{1/\alpha}}{n}$ with probability $1 - 3e^{-t}$ for any $t > 1$ where $C$ is a constant depending on $\alpha$.*

This is almost same as Corollary 1, but the exponential dependency on $R_2^L$ is replaced by the Lipschitz continuity $\kappa^2$.

*Proof of Corollary 2.* Suppose that

$$s_\ell \geq \min\left\{m_\ell, m_{\ell+1}, \lceil(4\kappa V_0 L)^{1/\alpha}\rceil\right\}, \tag{19}$$

then we show that Eq. (17) can be replaced by

$$\|f - f'\|_n \leq 4\sum_{\ell=1}^L V_0\kappa^2 s_\ell^{-\alpha} B_x, \tag{20}$$

where if $s_\ell = \min\{m_\ell, m_{\ell+1}\}$, then $s_\ell^{-\alpha}$ term can be replaced by 0 (which means no-compression in the layer $\ell$). Once we obtain this evaluations, then the following argument is same as the proof of Theorem 2 and Corollary 1 (Sec. B.3).

Let $\phi_\ell(x) = \eta \circ (W^{(\ell)}\eta(\cdot)) \circ \cdots \circ (W^{(1)}x)$ and $\phi_\ell^\sharp(x) = \eta \circ (W^{\sharp(\ell)}\eta(\cdot)) \circ \cdots \circ (W^{\sharp(1)}x)$ for $\ell = 1, \ldots, L-1$, and let $\phi_L(x) = G \circ (W^{(L)}\eta(\cdot)) \circ \cdots \circ (W^{(1)}x)$ and $\phi_L^\sharp(x) = G \circ (W^{\sharp(L)}\eta(\cdot)) \circ \cdots \circ (W^{\sharp(1)}x)$. Let $C_B := 2\kappa B_x$. We will show that

$$\|\phi_k - \phi_k^\sharp\|_n \leq 2\kappa V_0 C_B\left(\sum_{j=1}^k s_j^{-\alpha}\right), \quad \|\phi_k^\sharp\|_n \leq C_B,$$

for all $k = 1, \ldots, L$. We show this by inductive reasoning. To do so, we assume that, for $k = 1, \ldots, \ell-1$, this is satisfied, and then we show this for $k = \ell$. Note that, for all $k$ with $k < \ell$, it holds that, for any $\ell' > k$,

$$\begin{aligned}
\|M_{k,\ell'} \circ \phi_k^\sharp - M_{k-1,\ell'}\phi_{k-1}^\sharp\|_n &\leq \kappa(\|\phi_k^\sharp - \eta(W^{(k)}\phi_{k-1}^\sharp)\|_n + \tau\|\eta(W^{(k)}\phi_{k-1}^\sharp) - \phi_k\|_n) \\
&\leq \kappa(V_0 s_k^{-\alpha}\|\phi_{k-1}^\sharp\|_n + \tau\kappa\|\phi_{k-1}^\sharp - \phi_{k-1}\|_n) \\
&\leq \kappa\left(V_0 s_k^{-\alpha}C_B + \tau\kappa 2\kappa V_0 C_B\sum_{j\leq k-1} s_j^{-\alpha}\right) \quad \text{(by induction)} \\
&\leq \kappa V_0 C_B\left(s_k^{-\alpha} + \frac{1}{L}\sum_{j=1}^{k-1} s_j^{-\alpha}\right) \quad \text{(by the assumption of } \tau\text{)}.
\end{aligned}$$

Note that the term $s_k^{-\alpha}$ can be replaced by 0 if $s_k = \min\{m_k, m_{k+1}\}$ which corresponds to the full rank setting ($W^{\sharp(k)} = W^{(k)}$). Therefore, we have that

$$\|\phi_\ell - \phi_\ell^\sharp\|_n \leq \sum_{j=1}^\ell \|M_{j,\ell} \circ \phi_j^\sharp - M_{j-1,\ell} \circ \phi_{j-1}^\sharp\|_n \leq 2\kappa V_0 C_B\left(\sum_{k=1}^\ell s_k^{-\alpha}\right).$$

Under the setting (19), this gives that

$$\|\phi_\ell - \phi_\ell^\sharp\|_n \leq C_B/2.$$

Finally, noting that $\|\phi_\ell\|_n \leq C_B/2$, we have

$$\|\phi_\ell^\sharp\|_n \leq C_B.$$

This concludes the inductive reasoning.

Finally, noting that $f = \phi_L$ and $f' = \phi_L^\sharp$, we have Eq. (20). □

## B.5 Near low rank condition on the covariance matrix (Proof of Theorem 3 and Theorem 4)

Under Assumption 5, $\widehat{f}$ can be compressed as follows. Suppose that the network is compressed to smaller one upto the $\ell - 1$-th layer and the weight matrix of the compressed one is denoted by $(W^{\sharp(k)})_{k=1}^{\ell-1}$ where each $W^{\sharp(k)}$ has size $m_{k+1}^\sharp \times m_k^\sharp$ (here, $m_k^\sharp \leq m_k$ is assumed), and, in the $\ell - 1$-th layer, $W^{\sharp(\ell-1)}$ has size $m_\ell \times m_{\ell-1}^\sharp$. The input to the $\ell$-th layer of the compressed networkis denoted by $\phi_\ell^\sharp(x) = \eta(W^{\sharp(\ell-1)}\eta(\cdots W^{\sharp(1)}x)\cdots)$. Let $r_\ell^2 = \||\phi_\ell - \phi_\ell^\sharp|\|_n^2$ and $\Sigma_{(\ell)}^\sharp := \frac{1}{n}\sum_{i=1}^n \phi_\ell^\sharp(x_i)(\phi_\ell^\sharp(x_i))^\top$.

For a given matrix $\Sigma$ and a precision $r^2 > 0$, the degrees of freedom[3] are defined as

$$N_\ell(r^2, \Sigma) := \sum_{j=1}^{m_\ell} \frac{\sigma_j(\Sigma)}{\sigma_j(\Sigma) + r^2}. \tag{21}$$

Since the degrees of freedom are monotonically increasing with respect to each $\sigma_j(\Sigma)$, we can see that $N_\ell(r^2, \Sigma) \geq N_\ell(r^2, \Sigma')$ if $\Sigma \succeq \Sigma'$. Let[4]

$$m_\ell^\sharp = \lceil 5N_\ell(r^2, \Sigma_{(\ell)}^\sharp)\log(80N_\ell(r^2, \Sigma_{(\ell)}^\sharp))\rceil,$$

then Proposition 1 tells that there exits a matrix $\hat{A}_\ell \in m_\ell \times m_\ell^\sharp$ and $J_\ell \subset \{1, \ldots, m_\ell\}^{m_\ell^\sharp}$ such that

$$\|w^\top\phi_\ell^\sharp - w^\top\hat{A}_\ell\phi_{\ell,J_\ell}^\sharp\|_n^2 \leq 4r^2 w^\top\Sigma_{(\ell)}^\sharp(\Sigma_{(\ell)}^\sharp + r^2I)^{-1}w \leq 4r^2\|w\|^2, \tag{22}$$

for any $w \in \mathbb{R}^{m_\ell}$[5], and the norm of $\hat{A}_\ell$ is bounded as

$$\|\hat{A}_\ell\|_2 \leq \sqrt{\frac{20}{3}m_\ell}.$$

Next, we evaluate the degrees of freedom of $\Sigma_{(\ell)}^\sharp$. We bound this by using the degrees of freedom of $\widehat{\Sigma}_{(\ell)}$. First note that $r_\ell^2 = \||\phi_\ell - \phi_\ell^\sharp|\|_n^2$. Let $s \leq m$. For any matrix $U \in \mathbb{R}^{m_\ell \times s}$ such that $U^\top U = I_s$, $\text{Tr}[U^\top\Sigma_{(\ell)}^\sharp U] = P_n[\phi_\ell^{\sharp\top} UU^\top\phi_\ell^\sharp] \leq 2\{P_n[\phi_\ell^\top UU^\top\phi_\ell] + P_n[(\phi_\ell - \phi_\ell^\sharp)^\top UU^\top(\phi_\ell - \phi_\ell^\sharp)]\}$ by the Cauchy-Schwartz inequality. Here, let $U$ be the matrix that gives $P_n[\phi_\ell^\top UU^\top\phi_\ell] = \sum_{j=m_\ell-s+1}^{m_\ell} \sigma_j(\widehat{\Sigma}_{(\ell)}) = \inf_{U:U^\top U=I_s} P_n[\phi_\ell^\top UU^\top\phi_\ell]$, then by noticing $P_n[(\phi_\ell - \phi_\ell^\sharp)^\top UU^\top(\phi_\ell - \phi_\ell^\sharp)] \leq P_n\|\phi_\ell - \phi_\ell^\sharp\|^2 \leq r_\ell^2$, we obtain that $P_n[\phi_\ell^{\sharp\top} UU^\top\phi_\ell^\sharp] \leq 2[\sum_{j=m_\ell-s+1}^{m_\ell} \sigma_j(\widehat{\Sigma}_{(\ell)}) + r_\ell^2]$. Finally, by minimizing the left hand side with respect to $U$, we obtain that

$$\sum_{j=m_\ell-s+1}^{m_\ell} \sigma_j(\Sigma_{(\ell)}^\sharp) \leq 2\left(\sum_{j=m_\ell-s+1}^{m_\ell} \sigma_j(\widehat{\Sigma}_{(\ell)}) + r_\ell^2\right).$$

By setting $s = m_\ell - m + 1$ for $1 \leq m \leq m_\ell$, this indicates that

$$\sum_{j=m}^{m_\ell} \sigma_j(\Sigma_{(\ell)}^\sharp) \leq 2\left(\sum_{j=m}^{m_\ell} \sigma_j(\widehat{\Sigma}_{(\ell)}) + r_\ell^2\right) \leq 2\left(\sum_{j=m}^{m_\ell} \dot{\mu}_j^{(\ell)} + r_\ell^2\right).$$

---

[3]The definition is not dependent on $\ell$, but to make it clear that we are dealing with the $\ell$-th layer, we use the notation $N_\ell$.

[4]$\lceil x \rceil$ is the smallest integer that is not less than $x \in \mathbb{R}$.

[5]For a vector $x \in \mathbb{R}^m$ and index set $J \in \{1, \ldots, m\}^l$, $x_J$ is the vector corresponding to the index set $J$, that is, $x_J = (x_j)_{j\in J}$.

Now, let $\dot{m}_\ell := \min\{j \in \{1, \ldots, m_\ell\} \mid \dot{\mu}_j^{(\ell)} \leq r^2\}$ (if $\dot{\mu}_{m_\ell}^{(\ell)} > r^2$, then we set $\dot{m}_\ell = m_\ell$). Then,

$$N_\ell(r^2, \Sigma_{(\ell)}^\sharp) = \sum_{j=1}^{m_\ell} \frac{\sigma_j(\Sigma_{(\ell)}^\sharp)}{\sigma_j(\Sigma_{(\ell)}^\sharp) + r^2} \leq \dot{m}_\ell + \sum_{j > \dot{m}_\ell} \frac{\sigma_j(\Sigma_{(\ell)}^\sharp)}{\sigma_j(\Sigma_{(\ell)}^\sharp) + r^2}$$

$$\leq \dot{m}_\ell + \sum_{j > \dot{m}_\ell} \frac{\sigma_j(\Sigma_{(\ell)}^\sharp)}{r^2} \leq \dot{m}_\ell + \frac{2}{r^2}\left(r_\ell^2 + \sum_{j > \dot{m}_\ell} \sigma_j(\widehat{\Sigma}_{(\ell)})\right)$$

$$\leq \dot{m}_\ell + \frac{2}{r^2}\left(r_\ell^2 + U_0\frac{\dot{m}_\ell^{1-\beta}}{\beta - 1}\right) \leq \dot{m}_\ell + \frac{2}{r^2}\left(r_\ell^2 + \frac{\dot{m}_\ell r^2}{\beta - 1}\right)$$

$$\leq \dot{m}_\ell + \frac{2}{r^2}\left(r_\ell^2 + \frac{\dot{m}_\ell r^2}{\beta - 1}\right) = \frac{\beta + 1}{\beta - 1}\dot{m}_\ell + 2\frac{r_\ell^2}{r^2}.$$

Now, let

$$r^2 = \frac{1}{4}\tilde{r}_\ell^2,$$

then

$$N_\ell(r^2, \Sigma_{(\ell)}^\sharp) \leq \frac{\beta + 1}{\beta - 1}\dot{m}_\ell + 8\frac{r_\ell^2}{\tilde{r}_\ell^2}.$$

We define the right hand side as $m_\ell^\sharp$:

$$m_\ell^\sharp := \frac{\beta + 1}{\beta - 1}\dot{m}_\ell + 8\frac{r_\ell^2}{\tilde{r}_\ell^2}.$$

We have, by Eq. (22),

$$\sum_{j=1}^{m_{\ell+1}} \|\eta(W_{j,:}^{(\ell)}\phi_\ell^\sharp) - \eta(W_{j,:}^{(\ell)}\hat{A}_\ell\phi_{\ell,J_\ell}^\sharp)\|_n^2 \leq 4\sum_{j=1}^{m_{\ell+1}} \|W_{j,:}^{(\ell)}\|^2 r^2$$

$$\leq 4R_{\mathrm{F}}^2 \times \frac{1}{4}\tilde{r}_\ell^2 = R_{\mathrm{F}}^2\tilde{r}_\ell^2.$$

By the induction assumption, we also have that

$$\|\|\eta(W^{(\ell)}\phi_\ell^\sharp) - \phi_{\ell+1}\|\|_n^2 = \|\|\eta(W^{(\ell)}\phi_\ell^\sharp) - \eta(W^{(\ell)}\phi_\ell)\|\|_n^2 \leq \|W^{(\ell)}\|_2^2 r_\ell^2 \leq R_2^2 r_\ell^2.$$

Combining these inequalities, if we define

$$\phi_{\ell+1}^\sharp = \eta(W^{(\ell)}\hat{A}_\ell\phi_{\ell,J_\ell}^\sharp(x)),$$

and set $W^{\sharp(\ell)} = W^{(\ell)}\hat{A}_\ell$ and reset $W^{\sharp(\ell-1)} \leftarrow W_{J_\ell,:}^{\sharp(\ell-1)}$, then it holds that

$$\|\|\phi_{\ell+1} - \phi_{\ell+1}^\sharp\|\|_n \leq r_{\ell+1}$$

where we let

$$r_{\ell+1} = R_2 r_\ell + R_{\mathrm{F}}\tilde{r}_\ell.$$

Letting $r_0 = 0$, by an induction argument, we obtain

$$r_{\ell+1} \leq \sum_{k=1}^{\ell} R_2^{(\ell-k)} R_{\mathrm{F}}\tilde{r}_k.$$

Finally, we obtain

$$\|\widehat{f} - f^\sharp\|_n^2 \leq r_L^2 \leq \left[\sum_{k=1}^{L} R_2^{(L-k)} R_{\mathrm{F}}\tilde{r}_\ell\right]^2,$$

for a compressed network $f^\sharp$ that has width $\mathbf{m}^\sharp = (m_1^\sharp, \ldots, m_L^\sharp)$ with parameters $W^{\sharp(\ell)} = W_{J_{\ell+1},:}^{(\ell)}\hat{A}_\ell$. Note that

$$\|W^{\sharp(\ell)}\|_2 \leq \|W_{J_{\ell+1},:}^{(\ell)}\|_2\|\hat{A}_\ell\|_2 \leq R_2\sqrt{\frac{20}{3}m_\ell}, \quad \|W^{\sharp(\ell)}\|_{\mathrm{F}} \leq \|W_{J_{\ell+1},:}^{(\ell)}\|_{\mathrm{F}}\|\hat{A}_\ell\|_2 \leq R_{\mathrm{F}}\sqrt{\frac{20}{3}m_\ell}.$$

Therefore, if we set $\widehat{\mathcal{G}} = \mathrm{NN}(\mathbf{m}^{\sharp}, \sqrt{\frac{20}{3} \max_{\ell} m_{\ell}} R_2, \sqrt{\frac{20}{3} \max_{\ell} m_{\ell}} R_F)$, then there exists $\widehat{g} \in \widehat{\mathcal{G}}$ such that

$$\|\widehat{f} - \widehat{g}\|_n \leq \hat{r}$$

where

$$\hat{r}^2 = r_L^2.$$

Moreover, applying Lemma 5 to $\widehat{\mathcal{G}}$ and the Dudley integral yields

$$\bar{R}(\widehat{\mathcal{G}}) \leq CM \sqrt{L \frac{\sum_{\ell=1}^{L} m_{\ell}^{\sharp} m_{\ell+1}^{\sharp}}{n} \log(nL(R_2 \vee 1)(\max_{\ell} m_{\ell} + 1)^2)}.$$

This gives the assertion of Theorem 3.

Here, we consider a situation where $R_F^2 \tilde{r}_{\ell}^2 = c_0^2 \frac{r_{\ell}^2 R_2^2}{\ell}$ for some constant $c_0 > 0$. Then it holds that

$$r_{\ell+1} = R_2 \left(1 + \sqrt{\frac{c_0^2}{\ell}}\right) r_{\ell} = R_2^{\ell} \prod_{k=1}^{\ell} \left(1 + \sqrt{\frac{c_0^2}{k}}\right) r_1 \leq R_2^{\ell} \exp\left(c_0(2\sqrt{\ell} - 1)\right) r_1.$$

Therefore, by setting $C_L := (1 \vee R_2)^L \exp\left(c_0(2\sqrt{L} - 1)\right)$, it holds that $r_{\ell} \leq C_L r_1$ for $\ell = 1, \ldots, L$, in particular, we have

$$\hat{r} \leq C_L r_1.$$

In this situation, the degrees of freedom are bounded by

$$N_{\ell}(r^2, \Sigma_{(\ell)}^{\sharp}) \leq m_{\ell}^{\sharp} = \frac{\beta+1}{\beta-1} \dot{m}_{\ell} + 8\ell \frac{R_F^2}{c_0^2 R_2^2}.$$

Next, we bound $\dot{m}_{\ell}$. To do so, we should bound $\tilde{r}_{\ell}$ from below. Note that

$$\tilde{r}_{\ell} = \frac{c_0 R_2}{R_F} \frac{1}{\sqrt{\ell}} r_{\ell} = \frac{c_0 R_2}{R_F} \frac{1}{\sqrt{\ell}} R_2^{\ell-1} \prod_{k=1}^{\ell-1} \left(1 + \sqrt{\frac{c_0^2}{k}}\right) r_1 \geq \frac{c_0 R_2^{\ell}}{R_F} \frac{1}{\sqrt{\ell}} \prod_{k=1}^{\ell-1} \left(1 + \sqrt{\frac{c_0^2}{k}}\right) r_1$$

$$\geq \frac{c_0 R_2^{\ell}}{R_F} \frac{1}{\sqrt{\ell}} \left(1 + \sum_{k=1}^{\ell-1} \sqrt{\frac{c_0^2}{k}}\right) r_1 \geq \frac{c_0 R_2^{\ell}}{R_F} \frac{1}{\sqrt{\ell}} \left(1 + 2c_0(\sqrt{\ell} - 1)\right) r_1$$

$$\geq \frac{c_0 R_2^{\ell}}{R_F} \left(\frac{1}{\sqrt{\ell}} + 2c_0(1 - \frac{1}{\sqrt{\ell}})\right) r_1.$$

Hence,

$$\dot{m}_{\ell} \leq (\tilde{r}_{\ell}^2/(4U_0))^{-1/\beta} \leq (4U_0)^{1/\beta} \left\{\frac{c_0 R_2^{\ell}}{R_F} \left[\frac{1}{\sqrt{\ell}} + 2c_0(1 - \frac{1}{\sqrt{\ell}})\right] r_1\right\}^{-2/\beta}$$

$$\leq (4U_0)^{1/\beta} \left[\frac{R_F}{c_0 R_2^{\ell}}\right]^{2/\beta} \left(\frac{1}{2} \wedge c_0\right)^{-2/\beta} r_1^{-2/\beta} \leq \left[\frac{4U_0 R_F^2}{(0.5 \wedge c_0)^2 c_0^2 R_2^{2\ell}}\right]^{1/\beta} r_1^{-2/\beta}.$$

By Lemma 3, we can evaluate $r_*^2$ for $\widehat{\mathcal{F}}$ satisfying Assumption 4 as

$$r_*^2 \leq C \frac{(M+1)(S_1 + 1 + S_2 \log(n)) + Mt}{n} \vee M^{\frac{2\alpha-1}{2\alpha+1}} \left(\frac{S_3}{n}\right)^{\frac{2\alpha}{1+2\alpha}},$$

where

$$S_1 = \sum_{\ell=1}^{L} m_{\ell}^{\sharp} m_{\ell+1}^{\sharp},$$

$$S_2 = LS_1 \log(L(R_2 \vee 1)(\max_{\ell} m_{\ell} + 1)^2) = O\left(L \sum_{\ell=1}^{L} m_{\ell}^{\sharp} m_{\ell+1}^{\sharp} \log(n)\right),$$

$$S_3 = (\sum_{\ell=1}^{L} m_\ell)(2LV_0 R_2^{L-1} B_x)^{1/\alpha}[\log(n) + 2L\log(2L(R_2 \vee 1)(\max_\ell m_\ell + 1)^2)]$$

$$= O\left(L(\sum_{\ell=1}^{L} m_\ell)(2LV_0 R_2^{L-1} B_x)^{1/\alpha}\log(n)\right),$$

Then, the overall generalization error is upper bounded by

$$\Psi(\widehat{f}) \leq \widehat{\Psi}(\widehat{f}) + C\left[r_*^2 + \sqrt{\frac{S_3}{n}\hat{r}^{2(1-1/2\alpha)}} + \sqrt{(M^2 + \hat{r}^2)L\frac{\sum_{\ell=1}^{L} m_\ell^\sharp m_{\ell+1}^\sharp}{n}\log(n)} + \frac{1+Mt}{n}\right],$$

with probability $1 - 3e^{-t}$ for all $t \geq 1$. By letting $Q'_{L,\alpha,n} := L\frac{(2LV_0 R_2^{L-1} B_x)^{1/\alpha}\log(n)}{n}$ and assuming $\hat{r} \leq 1$, the second and third terms in $C[\cdot]$ is bounded by

$$\sqrt{Q'_{L,\alpha,n}(\sum_{\ell=1}^{L} m_\ell)(C_L r_1)^{2(1-1/2\alpha)}} + C'M\sqrt{L\frac{\sum_{\ell=1}^{L}[\dot{m}_\ell + \ell R_F^2/(c_0^2 R_2^2)]^2}{n}\log(n)^3}$$

$$\leq \sqrt{Q'_{L,\alpha,n}(\sum_{\ell=1}^{L} m_\ell)(C_L r_1)^{2(1-1/2\alpha)}} + 2C'M\sqrt{L\frac{L\left[\frac{4U_0 R_F^2}{(0.5\wedge c_0)^2 c_0^2(1\wedge R_2)^{2L}}\right]^{2/\beta} r_1^{-4/\beta} + L^3 R_F^4/R_2^4}{n}\log(n)^3}.$$

Hence, by setting $C_L r_1 = \left(\frac{\sum_{\ell=1}^{L} m_\ell}{L}\right)^{-\frac{1}{4/\beta+2(1-1/2\alpha)}}$ which balances the first and the second terms, then $\hat{r} \leq C_L r_1 \leq 1$ and the right hand side is bounded by

$$\sqrt{Q'_{L,\alpha,n}(\sum_{\ell=1}^{L} m_\ell)^{\frac{4/\beta}{4/\beta+2(1-1/2\alpha)}} L^{\frac{2(1-1/2\alpha)}{4/\beta+2(1-1/2\alpha)}}}$$

$$+ CM\sqrt{L\frac{L^{\frac{2(1-1/2\alpha)}{4/\beta+2(1-1/2\alpha)}}(\sum_{\ell=1}^{L} m_\ell)^{\frac{4/\beta}{4/\beta+2(1-1/2\alpha)}}(C_L)^{4/\beta}\left[\frac{4U_0 R_F^2}{(0.5\wedge c_0)^2 c_0^2(1\wedge R_2)^{2L}}\right]^{2/\beta}}{n}\log(n)^3}$$

$$+ CM\frac{R_F^2}{R_2^2}\sqrt{\frac{L^4}{n}\log(n)^3}.$$

Finally, by setting $c_0 = 1/4$, we obtain the assertion for

$$Q_L = \left[\frac{4U_0 R_F^2(1\vee R_2)^L \exp\left(c_0(2\sqrt{L}-1)\right)}{(0.5\wedge c_0)^2 c_0^2(1\wedge R_2)^{2L}}\right]^{2/\beta} \leq \left[\frac{4U_0 R_F^2(1\vee R_2)^L \exp\left(\frac{1}{4}(2\sqrt{L}-1)\right)}{(0.25)^4(1\wedge R_2)^{2L}}\right]^{2/\beta}.$$

This gives Theorem 4.

## B.6   IMPROVED BOUND OF THEOREM 4 WITH LIPSCHITZ CONTINUITY CONSTRAINT

Here, we again note that there appears $R_2^L$ in $P_L$ and $Q_L$ in the bound of Theorem 4. This is due to a rough evaluation of the interlayer Lipschitz continuity. We can reduce this exponential dependency under Assumption 6.

**Corollary 3.** *Assume Assumption 6 in addition to Assumptions 4 and 5, then the bound in Theorem 4 holds for the following redefined $P_L$ and $Q_L$:*

$$P_L = (2LV_0\kappa^2 B_x)^{1/\alpha}, \quad Q_L = \left[\frac{4U_0 R_F^2 \exp\left(\frac{1}{4}(2\sqrt{L}-1)\right)}{(0.25)^4}\right]^{2/\beta},$$

*except that the term $M\frac{R_F^2 L^2}{R_2^2}\sqrt{\frac{\log(n)^3}{n}}$ is replaced by $M\kappa^2 R_F^2 L^2\sqrt{\frac{\log(n)^3}{n}}$:*

$$\Psi(\widehat{f}) \leq \widehat{\Psi}(\widehat{f}) + C\left[M\sqrt{\frac{[P_L \vee Q_L]L^{1+\frac{\beta}{(2\alpha-1)}+\beta}}{n}}\left(\sum_{\ell=1}^{L} m_\ell\right)^{\frac{4/\beta}{4/\beta+2(1-1/2\alpha)}}\log(n)^3\right.$$

$$+ M^{\frac{2\alpha-1}{2\alpha+1}} \left( LP_L \frac{\sum_{\ell=1}^L m_\ell}{n} \log(n) \right)^{\frac{2\alpha}{2\alpha+1}} + M\kappa^2 R_{\mathrm{F}}^2 L^2 \sqrt{\frac{\log(n)^3}{n}} + \frac{1+Mt}{n} \bigg].$$

*Proof of Corollary 3.* To show Corollary 3, we set

$$\tilde{r}_\ell = \frac{1}{\sqrt{\ell}} \prod_{k=1}^{\ell-1} \left( 1 + \sqrt{\frac{c_0^2}{k}} \right) \frac{r_1}{R_{\mathrm{F}}},$$

where $c_0$ is a constant, and by the same argument as in the proof of Corollary 2, we can show that

$$r_\ell \leq 2\kappa \sum_{k=1}^{\ell} \frac{1}{\sqrt{k}} \prod_{j=1}^{k-1} \left( 1 + \sqrt{\frac{c_0^2}{j}} \right) r_1.$$

Then, through a cumbersome calculation, we have that

$$\frac{r_\ell}{\tilde{r}_\ell} \leq C \frac{\kappa R_{\mathrm{F}}}{c_0} \sqrt{\ell},$$

for a universal constant $C$. Moreover, we can show that $r_\ell$ can be bounded as

$$r_\ell \leq 2\kappa \frac{e^{c_0(2\sqrt{\ell+1}-1)} - e^{c_0}}{c_0} r_1.$$

This also gives

$$r_L \leq C' \frac{\kappa}{c_0} \exp(c_0(2\sqrt{L} - 1)) r_1,$$

for a universal constant $C'$. Then, redefining $C_L = \kappa \exp(c_0(2\sqrt{L} - 1))$, we can apply the same argument as in the proof of Corollary 2. Indeed, we can show

$$\dot{m}_\ell \lesssim \left[ \frac{4U_0 R_{\mathrm{F}}^2}{(0.5 \wedge c_0)^2 c_0^2 \kappa^2} \right]^{1/\beta}, \quad m_\ell^\sharp = \frac{\beta+1}{\beta-1} \dot{m}_\ell + C^2 \ell \frac{R_{\mathrm{F}}^2}{c_0^2} r_1^{-2/\beta}.$$

From the above argument, if we set $\widehat{\mathcal{G}} = \mathrm{NN}(\mathbf{m}^\sharp, \sqrt{\frac{20}{3} \max_\ell m_\ell} R_2, \sqrt{\frac{20}{3} \max_\ell m_\ell} R_F)$, then there exists $\widehat{g} \in \widehat{\mathcal{G}}$ such that

$$\|\widehat{f} - \widehat{g}\|_n \leq \hat{r}$$

where

$$\hat{r}^2 = r_L^2.$$

Moreover, we can show

$$r_*^2 \leq C \frac{(M+1)(S_1 + 1 + S_2 \log(n)) + Mt}{n} \vee M^{\frac{2\alpha-1}{2\alpha+1}} \left( \frac{S_3}{n} \right)^{\frac{2\alpha}{1+2\alpha}},$$

where

$$S_1 = \sum_{\ell=1}^L m_\ell^\sharp m_{\ell+1}^\sharp,$$

$$S_2 = LS_1 \log(L(R_2 \vee 1)(\max_\ell m_\ell + 1)^2) = O \left( L \sum_{\ell=1}^L m_\ell^\sharp m_{\ell+1}^\sharp \log(n) \right),$$

$$S_3 = (\sum_{\ell=1}^L m_\ell)(2LV_0\kappa^2 B_x)^{1/\alpha} [\log(n) + 2L \log(2L(R_2 \vee 1)(\max_\ell m_\ell + 1)^2)]$$

$$= O \left( L(\sum_{\ell=1}^L m_\ell)(2LV_0\kappa^2 B_x)^{1/\alpha} \log(n) \right).$$

Here, to evaluate $S_3$, we used the argument in Sec. B.4 (proof of Corollary 2).

The remaining argument is the same as the proof of Theorem 3 and Theorem 4 (Sec. B.5). $\qquad\square$

## C    AUXILIARY LEMMAS

In this section, we give several auxiliary lemmas that are used in the proof of the theorems. These results are not new at all, but we explicitly present them for completeness.

### C.1    COVERING NUMBER OF DEEP NETWORK MODELS

Define the neural network with height $L$, width $m$, sparsity constraint $S$ and norm constraint $B$ as

$$\Phi(L, m, S, B) := \{G \circ (W^{(L)}\eta(\cdot) + b^{(L)}) \circ \cdots \circ (W^{(1)}x + b^{(1)}) \mid W^{(L)} \in \mathbb{R}^{1 \times m}, \ b^{(L)} \in \mathbb{R},$$

$$W^{(1)} \in \mathbb{R}^{m \times d}, \ b^{(1)} \in \mathbb{R}^m, \ W^{(\ell)} \in \mathbb{R}^{m \times m}, \ b^{(\ell)} \in \mathbb{R}^m (1 < \ell < L),$$

$$\sum_{\ell=1}^{L}(\|W^{(\ell)}\|_0 + \|b^{(\ell)}\|_0) \le S, \max_{\ell} \|W^{(\ell)}\|_\infty \vee \|b^{(\ell)}\|_\infty \le B\},$$

where $\|\cdot\|_0$ is the $\ell_0$-norm of the matrix (the number of non-zero elements of the matrix) and $\|\cdot\|_\infty$ is the $\ell_\infty$-norm of the matrix (maximum of the absolute values of the elements).

The following evaluation of the covering number of the model $\Phi(L, m, S, B)$ is shown by Schmidt-Hieber (2019); Suzuki (2019).

**Lemma 4** (Covering number evaluation (Schmidt-Hieber, 2019; Suzuki, 2019)). *The covering number of $\Phi(L, m, S, B)$ can be bounded by*

$$\log \mathcal{N}(\Phi(L, m, S, B), \|\cdot\|_\infty, \delta) \le S \log(\delta^{-1}L(B \vee 1)^{L-1}(m+1)^{2L})$$

$$\le 2SL \log((B \vee 1)(m+1)) + S \log(\delta^{-1}L).$$

*Proof of Lemma 4.* Given a network $f \in \Phi(L, m, S, B)$ expressed as

$$f(x) = G \circ (W^{(L)}\eta(\cdot) + b^{(L)}) \circ \cdots \circ (W^{(1)}x + b^{(1)}),$$

let

$$\mathcal{A}_k(f)(x) = \eta \circ (W^{(k-1)}\eta(\cdot) + b^{(k-1)}) \circ \cdots \circ (W^{(1)}x + b^{(1)}),$$

and

$$\mathcal{B}_k(f)(x) = G \circ (W^{(L)}\eta(\cdot) + b^{(L)}) \circ \cdots \circ (W^{(k)}\eta(x) + b^{(k)}),$$

for $k = 2, \ldots, L$. Corresponding to the last and first layer, we define $\mathcal{B}_{L+1}(f)(x) = x$ and $\mathcal{A}_1(f)(x) = x$. Then, it is easy to see that $f(x) = \mathcal{B}_{k+1}(f) \circ (W^{(k)} \cdot + b^{(k)}) \circ \mathcal{A}_k(f)(x)$. Now, suppose that a pair of different two networks $f, g \in \Phi(L, m, S, B)$ given by

$$f(x) = G \circ (W^{(L)}\eta(\cdot) + b^{(L)}) \circ \cdots \circ (W^{(1)}x + b^{(1)}), \ g(x) = G \circ (W^{(L)'}\eta(\cdot) + b^{(L)'}) \circ \cdots \circ (W^{(1)'}x + b^{(1)'}),$$

has a parameters with distance $\delta$: $\|W^{(\ell)} - W^{(\ell)'}\|_\infty \le \delta$ and $\|b^{(\ell)} - b^{(\ell)'}\|_\infty \le \delta$. Now, not that $\|\mathcal{A}_k(f)\|_\infty \le \max_j \|W_{j,:}^{(k-1)}\|_1\|\mathcal{A}_{k-1}(f)\|_\infty + \|b^{(k-1)}\|_\infty \le mB\|\mathcal{A}_{k-1}(f)\|_\infty + B \le (B \vee 1)(m+1)\|\mathcal{A}_{k-1}(f)\|_\infty \le (B \vee 1)^{k-1}(m+1)^{k-1}$, and similarly the Lipshitz continuity of $\mathcal{B}_k(f)$ with respect to $\|\cdot\|_\infty$-norm is bounded as $(Bm)^{L-k+1}$. Then, it holds that

$$|f(x) - g(x)|$$

$$= \left| \sum_{k=1}^{L} \mathcal{B}_{k+1}(g) \circ (W^{(k)} \cdot + b^{(k)}) \circ \mathcal{A}_k(f)(x) - \mathcal{B}_{k+1}(g) \circ (W^{(k)'} \cdot + b^{(k)'}) \circ \mathcal{A}_k(f)(x) \right|$$

$$\le \sum_{k=1}^{L}(Bm)^{L-k}\|(W^{(k)} \cdot + b^{(k)}) \circ \mathcal{A}_k(f)(x) - (W^{(k)'} \cdot + b^{(k)'}) \circ \mathcal{A}_k(f)(x)\|_\infty$$

$$\le \sum_{k=1}^{L}(Bm)^{L-k}\delta[m(B \vee 1)^{k-1}m^{k-1} + 1]$$

$$\le \sum_{k=1}^{L}(Bm)^{L-k}\delta(B \vee 1)^{k-1}m^k \le \delta L(B \vee 1)^{L-1}(m+1)^L.$$

Thus, for a fixed sparsity pattern (the locations of non-zero parameters), the covering number is bounded by $\left(\delta/[L(B \vee 1)^{L-1}(m+1)^L]\right)^{-S}$. There are the number of configurations of the sparsity pattern is bounded by $\binom{(m+1)^L}{S} \leq (m+1)^{LS}$. Thus, the covering number of the whole space $\Phi$ is bounded as

$$(m+1)^{LS}\left\{\delta/[L(B \vee 1)^{L-1}(m+1)^L]\right\}^{-S} = [\delta^{-1}L(B \vee 1)^{L-1}(m+1)^{2L}]^S,$$

which gives the assertion.

$\square$

**Lemma 5** (Covering number evaluation). *Let* $\mathrm{NN}(\mathbf{m}, R_2, R_\mathrm{F})$ *be the set of neural networks with depth* $\mathbf{m}(m_1,\ldots,m_L)$, $\|W^{(\ell)}\|_\infty \leq R_2$ *and* $\|W^{(\ell)}\|_\mathrm{F} \leq R_\mathrm{F}$. *The covering number of* $\mathrm{NN}(\mathbf{m}, R_2, R_\mathrm{F})$ *can be bounded by*

$$\log \mathcal{N}(\mathrm{NN}(\mathbf{m}, R_2, R_\mathrm{F}), \|\cdot\|_\infty, \delta)$$
$$\leq (\sum_{\ell=1}^L m_\ell m_{\ell+1}) \log(\delta^{-1} L (R_2 \vee 1)^{L-1}(\max_\ell m_\ell + 1)^L)$$
$$\leq (\sum_{\ell=1}^L m_\ell m_{\ell+1}) \log(\delta^{-1}) + L(\sum_{\ell=1}^L m_\ell m_{\ell+1}) \log(L(R_2 \vee 1)(\max_\ell m_\ell + 1)).$$

*Moreover, the set of networks with low rank weight matrices,* $\mathrm{NN}(\mathbf{m}, \mathbf{s}, R_2, R_\mathrm{F})$, *has the following covering number bound:*

$$\log \mathcal{N}(\mathrm{NN}(\mathbf{m}, \mathbf{s}, R_2, R_\mathrm{F}), \|\cdot\|_\infty, \delta)$$
$$\leq \sum_{\ell=1}^L s_\ell(m_\ell + m_{\ell+1}) \log(\delta^{-1} L (R_2 \vee 1)^{2L-1}(\max_\ell m_\ell + 1)^{2L}).$$

*Proof of Lemma 5.* Let $B = R_2$, $m = \max_\ell m_\ell$, and $S = \sum_{\ell=1}^L m_\ell m_{\ell+1}$, then we can see that $\mathrm{NN}(\mathbf{m}, R_2, R_\mathrm{F})$ is a subset of $\Phi(L, m, S, B)$ because $\|W\|_\infty \leq \|W\|_2$. Hence Lemma 4 gives the first assertion. As for the second one, we can easily check that the covering number of $\mathrm{NN}(\mathbf{m}, \mathbf{s}, R_2, R_\mathrm{F})$ can be bounded by the one given in Lemma 5 for $\Phi(2L, m, S, B)$ with $S = \sum_{\ell=1}^L s_\ell(m_\ell + m_{\ell+1})$. Then, we obtain the second assertion.

$\square$

## C.2 Compression error bound for one layer

The following proposition was shown by Bach (2017); Suzuki et al. (2018). Let $\widehat{\Sigma}_{I,I'} \in \mathbb{R}^{K \times H}$ for integers $K, H \in \mathbb{N}$ and a matrix $\widehat{\Sigma}_{I,I'} \in \mathbb{R}^{K \times H}$ be a matrix $(\widehat{\Sigma}_{i,j})_{i \in I, j \in I'}$ for the index sets $I \in [m_\ell]^K$ and $I' \in [m_\ell]^H$. Let $F = \{1, \ldots, m_\ell\}$ be the full index set. Let the degrees of freedom corresponding to $\widehat{\Sigma}$ be $\hat{N}(\lambda) := N_\ell(\lambda, \widehat{\Sigma})$ (see Eq. (21)) for $\lambda > 0$.

**Proposition 1.** *Let* $U = (U_{j,l})_{j,l}$ *is the orthogonal matrix that diagonalizes* $\widehat{\Sigma}$, *that is,* $\widehat{\Sigma} = U\mathrm{diag}(\hat{\mu}_1, \ldots, \hat{\mu}_{m_\ell}) U^\top$ *for* $\hat{\mu}_j \geq 0$ $(j = 1, \ldots, m_\ell)$. *Define*

$$\tau'_j = \frac{1}{\hat{N}(\lambda)} \sum_{l=1}^{m_\ell} U_{j,l}^2 \frac{\hat{\mu}_l^{(\ell)}}{\hat{\mu}_l^{(\ell)} + \lambda} = \frac{1}{\hat{N}(\lambda)} [\widehat{\Sigma}(\widehat{\Sigma} + \lambda \mathrm{I})^{-1}]_{j,j} \quad (j \in \{1, \ldots, m_\ell\}). \tag{23}$$

*For* $\lambda > 0$, *if*

$$m \geq 5\hat{N}(\lambda) \log(80\hat{N}(\lambda)),$$

*then there exist* $v_1, \ldots, v_m \in \{1, \ldots, m_\ell\}$ *such that, for every* $\alpha \in \mathbb{R}^{m_\ell}$,

$$\inf_{\beta \in \mathbb{R}^m} \left\{ \left\| \alpha^\top \eta(\hat{F}_{\ell-1}(\cdot)) - \sum_{j=1}^m \beta_j \eta(\hat{F}_{\ell-1}(\cdot))_{v_j} \right\|_n^2 + m\lambda \|\beta\|_{\tau'}^2 \right\} \leq 4\lambda \alpha^\top \widehat{\Sigma}(\widehat{\Sigma} + \lambda \mathrm{I})^{-1}\alpha, \tag{24}$$

and $\sum_{j=1}^{m} \tau_j'^{-1} \leq \frac{5}{3} m \times m_\ell$, where $\|\beta\|_{\tau'}^2 := \sum_{j=1}^{m} \beta_j^2 \tau_j'$. Let $\tau := m\lambda\tau'$ and $I_\tau = \text{diag}(\tau)$. Then, $\hat{A} := \widehat{\Sigma}_{F,J}(\widehat{\Sigma}_{J,J} + I_\tau)^{-1}$ for $J = \{v_1, \ldots, v_m\}$ satisfies

$$\|\hat{A}\|_2 \leq \sqrt{\frac{20}{3} m_\ell},$$

and the optimal $\beta$ that achieves the infimum is given by $\hat{\beta} = \hat{A}^\top \alpha$ for any $\alpha \in \mathbb{R}^{m_\ell}$.

## C.3 CONCENTRATION INEQUALITY

**Proposition 2** (Talagrand's Concentration Inequality (Talagrand, 1996; Bousquet, 2002))**.** *Let $\mathcal{G}$ be a function class on $\mathcal{X}$ that is separable with respect to $\infty$-norm, and $\{x_i\}_{i=1}^{n}$ be i.i.d. random variables with values in $\mathcal{X}$. Furthermore, let $B \geq 0$ and $U \geq 0$ be $B := \sup_{g \in \mathcal{G}} \mathrm{E}[(g - \mathrm{E}[g])^2]$ and $U := \sup_{g \in \mathcal{G}} \|g\|_\infty$, then for $Z := \sup_{g \in \mathcal{G}} \left| \frac{1}{n} \sum_{i=1}^{n} g(x_i) - \mathrm{E}[g] \right|$, we have*

$$P\left( Z \geq 2\mathrm{E}[Z] + \sqrt{\frac{2Bt}{n}} + \frac{2Ut}{n} \right) \leq e^{-t}, \tag{25}$$

*for all $t > 0$.*

## D NUMERICAL EXPERIMENTS

In this section, we experimentally validate the assumptions we made in the theoretical analysis and investigate how large the intrinsic dimensionality becomes. We use VGG-19 network trained on CIFAR-10. The VGG-19 network have 16 convolution layers (named c0, ..., c15) and 3 fully connected layers (named l16, ..., l18). The size of each filter in each convolution layer is $3 \times 3$. Our theory does not support a convolution layer in a strict sense, but we adopt it as follows. If the convolution layer in the $\ell$-th layer has the input channel size $m_\ell$ and the output channel size $m_{\ell+1}$ with the filter size $k \times k$ (in our case $k = 3$), then the weight matrix is given as a 4-way tensor with the size $m_{\ell+1} \times m_\ell \times k \times k$: $W^{(\ell)} \in \mathbb{R}^{m_{\ell+1} \times m_\ell \times k \times k}$. Although a singular value of a 4-way tensor is not well-defined, we can perform a low rank approximation of the weight matrix by folding out the tensor to a large matrix. Actually, considering "similarity" between the filters as

$$K_{(\ell)} := \left( \sum_{c=1}^{m_\ell} \sum_{\kappa_1, \kappa_2 = 1}^{k,k} W^{(\ell)}_{i,c,\kappa_1,\kappa_2} W^{(\ell)}_{j,c,\kappa_1,\kappa_2} \right)_{i,j=1}^{m_{\ell+1}} \in \mathbb{R}^{m_{\ell+1} \times m_{\ell+1}},$$

then we can easily see that, for $s_\ell \in [m_{\ell+1}]$, it holds that

$$\|W^{(\ell)} - P^\top W'\|_{\mathrm{F}} = \sqrt{\sum_{j'=\dot{m}}^{m_{\ell+1}} \sigma_{j'}(K_{(\ell)})},$$

where $P \in \mathbb{R}^{s_\ell \times m_{\ell+1}}$ is a projection matrix to the eigen-space corresponding to the $s_\ell$ largest singular values of $K_{(\ell)}$, $W' := PW^{(\ell)} = (\sum_{i'=1}^{m_{\ell+1}} P_{i,i'} W^{(\ell)}_{i',j,k,k'})_{i,j,k,k' \in [s_\ell] \times [m_\ell] \times [k] \times [k]}$ and the Frobenius norm $\| \cdot \|_{\mathrm{F}}$ of a tensor is the Euclidean norm as a vector. Therefore, we can use the eigenvalues of $K_{(\ell)}$ to evaluate the redundancy of parameters among filters.

As for the covariance matrix in a convolution layer, we also apply the same argument. That is, the input to the $\ell$-th layer (which is a convolution layer) is given by $\phi_\ell(x) \in \mathbb{R}^{m_\ell \times I \times J}$ where $I$ and $J$ are the width and height of the input, and we define the following "covariance" matrix as a similarity measure between the channels:

$$\widehat{\Sigma}_{(\ell)} = \left( \frac{1}{n} \sum_{i'=1}^{n} \sum_{1 \leq c_1 \leq I} \sum_{1 \leq c_2 \leq J} \phi_{\ell,(i,c_1,c_2)}(x_{i'}) \phi_{\ell,(j,c_1,c_2)}(x_{i'}) \right)_{i,j=1}^{m_\ell, m_\ell} \in \mathbb{R}^{m_\ell \times m_\ell}.$$

This also serve the redundancy measure and analogous argument to the main text can be applied.

**Near low rank properties of the covariance matrix and weight matrix**   Here, we see plausibility of the near low rank assumptions we made in the analysis. Figure 1 presents the eigenvalues of covariance matrix $\widehat{\Sigma}_{(\ell)}$ in each of layer c2, c7, c12 and l16. The eigenvalues are sorted in decreasing order. We can see that the eigenvalue distributions are highly concentrated around 0 and the eigenvalues decrease quickly, which indicates the near low rank property of $\widehat{\Sigma}_{(\ell)}$.

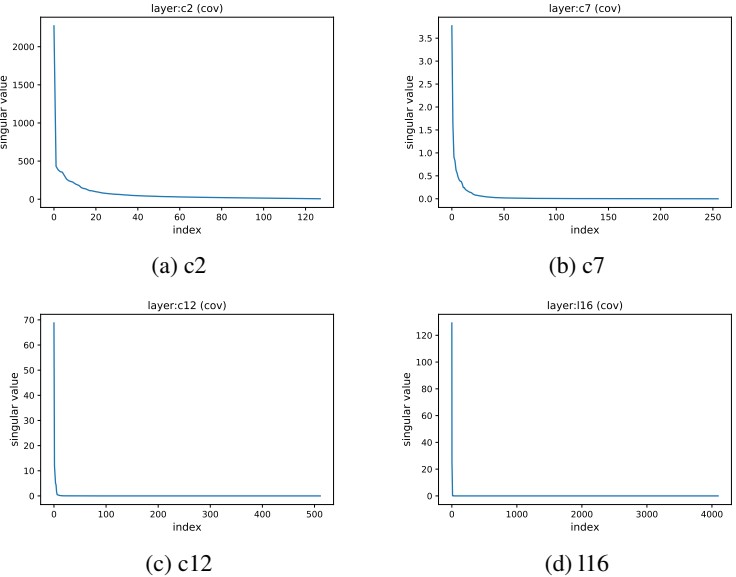

Figure 1: Eigenvalue distribution of the covariance matrices

Next, we plot the eigenvalues of $K_{(\ell)}$ in Figure 2 for layer c2, c7, c12 and l16. We again observe a rapid decrease of the eigenvalues. These results justify our theoretical assumptions.

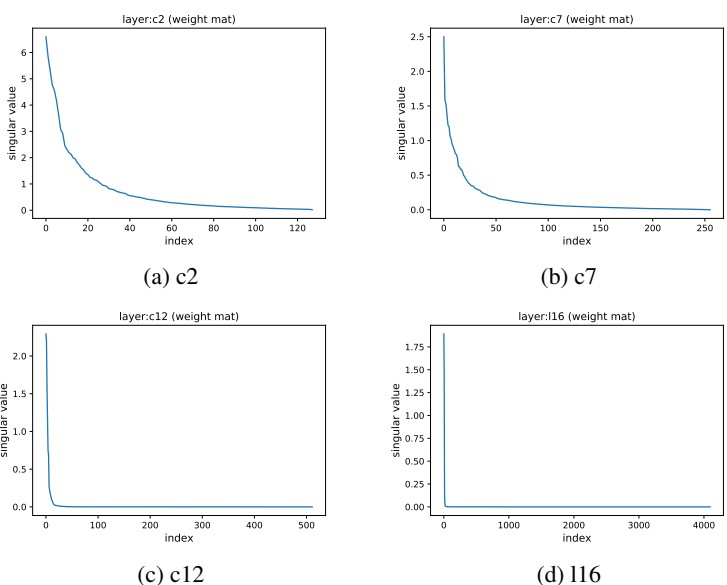

Figure 2: Singular-value distribution of the weight matrices

**Intrinsic dimensionality**   Here, we calculate the intrinsic dimensionalities of the VGG-19 network. For that purpose, we set a threshold parameter $\nu \in \{10^{-1}, 10^{-2}, 10^{-3}\}$ and compute

Table 3: The effective ranks $\dot{m}_\ell$ and $s_\ell$ in each layer for each threshold $\nu$. "In/Out" indicates the channel sizes of the input and output.

| layer | In/Out | Cov ($\dot{m}_\ell$) | | | Weight ($s_\ell$) | | |
|---|---|---|---|---|---|---|---|
| | | $\nu : 10^{-1}$ | $10^{-2}$ | $10^{-3}$ | $\nu : 10^{-1}$ | $10^{-2}$ | $10^{-3}$ |
| c0 | 3/64 | 3 | 3 | 3 | 9 | 16 | 19 |
| c1 | 64/64 | 5 | 21 | 51 | 26 | 62 | 64 |
| c2 | 64/128 | 5 | 33 | 64 | 37 | 110 | 128 |
| c3 | 128/128 | 9 | 76 | 128 | 50 | 128 | 128 |
| c4 | 128/256 | 11 | 110 | 128 | 102 | 255 | 256 |
| c5 | 256/256 | 13 | 200 | 256 | 93 | 256 | 256 |
| c6 | 256/256 | 17 | 219 | 256 | 55 | 230 | 256 |
| c7 | 256/256 | 11 | 110 | 253 | 37 | 175 | 252 |
| c8 | 256/512 | 10 | 35 | 129 | 30 | 122 | 349 |
| c9 | 512/512 | 6 | 33 | 79 | 16 | 60 | 217 |
| c10 | 512/512 | 3 | 15 | 41 | 16 | 42 | 127 |
| c11 | 512/512 | 2 | 12 | 23 | 11 | 39 | 129 |
| c12 | 512/512 | 2 | 7 | 16 | 7 | 18 | 46 |
| c13 | 512/512 | 3 | 6 | 16 | 9 | 22 | 46 |
| c14 | 512/512 | 3 | 7 | 18 | 16 | 38 | 59 |
| c15 | 512/512 | 4 | 14 | 28 | 36 | 51 | 92 |
| l16 | 512/4096 | 5 | 10 | 11 | 11 | 21 | 49 |
| l17 | 4096/4096 | 7 | 10 | 11 | 10 | 49 | 990 |
| l18 | 4096/10 | 6 | 10 | 10 | 9 | 9 | 9 |

$s_\ell := \#\{j \in [m_{\ell+1}] \mid \sigma_j(K_{(\ell)}) \geq \nu \times \max_{j'} \sigma_{j'}(K_{(\ell)})\}$ and $\dot{m}_\ell := \#\{j \in [m_\ell] \mid \sigma_j(\widehat{\Sigma}_{(\ell)}) \geq \nu \times \max_{j'} \sigma_{j'}(\widehat{\Sigma}_{(\ell)})\}$ (which corresponds to setting $\tilde{r}_\ell^2 = 4\nu \times \max_{j'} \sigma_{j'}(\widehat{\Sigma}_{(\ell)})$). Table 4 summarizes the effective ranks $\dot{m}_\ell, s_\ell$ of all layers. We can see that the effective ranks can be much smaller than the channel sizes in several layers (especially layers from c8 to l18,), which indicates the network has high redundancy and its intrinsic dimensionality could be much smaller than the actual number of parameters.

Next, we compute the intrinsic dimensionality of each layer based on the effective ranks calculated above. Basically, the main term of our bound (Theorem 4) is given by $\sqrt{\sum_{\ell=1}^L \frac{\dot{m}_{\ell+1}\dot{m}_\ell}{n}}$ (note that $m_\ell^\sharp$ is essentially controlled by $\dot{m}_\ell$). Hence, we employ $\dot{m}_{\ell+1} \times \dot{m}_\ell$ as the intrinsic dimensionality of each fully connected layer. As for a convolution layer, we employ $\dot{m}_{\ell+1}\dot{m}_\ell k^2$ as the intrinsic dimensionality which is the number of parameters of compressed network. We also calculate the intrinsic dimensionality obtained by compressing only the weight matrix. That is given by $s_\ell m_\ell k^2 + m_{\ell+1}s_\ell$. Both of them are summarized in Table 4. We can see that the intrinsic dimensionality is smaller than the actual number of parameters. In particular, it is much smaller for higher layers such as c8 to l18. This indicates that the information required for classification is almost distilled in the first few layers and the contribution of the subsequent layers would be much smaller than the earlier layers. Moreover, we see that compressing the network using the covariance matrix gives smaller intrinsic dimensionalities than the weight matrix. This is because the improvement induced by decreasing $\dot{m}_\ell$ is a quadratic order but that by $s_\ell$ is just a linear order. Another reason is that the effective rank of the covariance matrix is more data dependent in a sense that it is strongly dependent on the distribution of the data, and thus it can capture data dependent redundancy more efficiently (see also Figures 1 and 2). Since the intrinsic dimensionality is much smaller than the actual number of parameters, the VC-dimension bound is too pessimistic and a compression based bound like ours gives a better generalization error bound.

Finally, we give a comparison of intrinsic dimensionalities calculated by Arora et al. (2018) and ours. We borrowed the values presented in the paper (Arora et al., 2018). We would like to note that the comparison is not completely fair because the intrinsic dimensionality of both our analysis and that of Arora et al. (2018) neglect constant functors (such as depth), and thus the final generalization error is not merely determined by the raw values. However, the comparison offers better understanding of our analysis by observing difference and similarity between them. We can see

Table 4: The intrinsic dimensionality in each layer for each threshold $\nu$. Here again, "In/Out" indicates the channel sizes of the input and output. "Orig" indicates the number of parameters ($m_{\ell+1}m_\ell \times$ filter size) in each layer.

| layer | In/Out | Orig | Cov | | | Weight | | |
|---|---|---|---|---|---|---|---|---|
| | | | $\nu : 10^{-1}$ | $10^{-2}$ | $10^{-3}$ | $\nu : 10^{-1}$ | $10^{-2}$ | $10^{-3}$ |
| c0 | 3/64 | 1,728 | 135 | 567 | 1,377 | 910 | 910 | 910 |
| c1 | 64/64 | 36,864 | 225 | 6,237 | 29,376 | 1,920 | 1,920 | 1,920 |
| c2 | 64/128 | 73,728 | 405 | 22,572 | 73,728 | 6,336 | 11,264 | 13,376 |
| c3 | 128/128 | 147,456 | 891 | 75,240 | 147,456 | 33,280 | 79,360 | 81,920 |
| c4 | 128/256 | 294,912 | 1,287 | 198,000 | 294,912 | 52,096 | 154,880 | 180,224 |
| c5 | 256/256 | 589,824 | 1,989 | 394,200 | 589,824 | 128,000 | 327,680 | 327,680 |
| c6 | 256/256 | 589,824 | 1,683 | 216,810 | 582,912 | 261,120 | 652,800 | 655,360 |
| c7 | 256/256 | 589,824 | 990 | 34,650 | 293,733 | 238,080 | 655,360 | 655,360 |
| c8 | 256/512 | 1,179,648 | 540 | 10,395 | 91,719 | 154,880 | 647,680 | 720,896 |
| c9 | 512/512 | 2,359,296 | 162 | 4,455 | 29,151 | 189,440 | 896,000 | 1,290,240 |
| c10 | 512/512 | 2,359,296 | 54 | 1,620 | 8,487 | 153,600 | 624,640 | 1,786,880 |
| c11 | 512/512 | 2,359,296 | 36 | 756 | 3,312 | 81,920 | 307,200 | 1,111,040 |
| c12 | 512/512 | 2,359,296 | 54 | 378 | 2,304 | 81,920 | 215,040 | 650,240 |
| c13 | 512/512 | 2,359,296 | 81 | 378 | 2,592 | 56,320 | 199,680 | 660,480 |
| c14 | 512/512 | 2,359,296 | 108 | 882 | 4,536 | 35,840 | 92,160 | 235,520 |
| c15 | 512/512 | 2,359,296 | 180 | 1,260 | 2,772 | 46,080 | 112,640 | 235,520 |
| l16 | 512/4096 | 2,097,152 | 35 | 100 | 121 | 73,728 | 175,104 | 271,872 |
| l17 | 4096/4096 | 16,777,216 | 42 | 100 | 110 | 294,912 | 417,792 | 753,664 |
| l18 | 4096/10 | 40,960 | 42 | 90 | 90 | 45,166 | 86,226 | 201,194 |

Table 5: Comparison of the intrinsic dimensionality of our analysis and that in Arora et al. (2018).

| layer | In/Out | Orig | Arora et al. (2018) | Cov | | |
|---|---|---|---|---|---|---|
| | | | | $\nu : 10^{-1}$ | $10^{-2}$ | $10^{-3}$ |
| c0 | 3/64 | 1,728 | 1,645 | 135 | 567 | 1,377 |
| c3 | 128/128 | 147,456 | 644,654 | 891 | 75,240 | 147,456 |
| c5 | 256/256 | 589,824 | 3,457,882 | 1,989 | 394,200 | 589,824 |
| c8 | 256/512 | 1,179,648 | 36,920 | 540 | 10,395 | 91,719 |
| c11 | 512/512 | 2,359,296 | 22,735 | 36 | 756 | 3,312 |
| c14 | 512/512 | 2,359,296 | 26,584 | 108 | 882 | 4,536 |

that our intrinsic dimensionality gives a smaller number than theirs. This is because compression through the covariance matrix gives quadratic factor improvement while compression through low rank property of the weight matrices gives linear order improvement. This indicates considering near low rank properties of both of weight matrices and covariance matrices yields sharper bounds. It can be realized by our unified theoretical frame-work.

