# OpenReview forum: "Compression based bound for non-compressed network: unified generalization error analysis of large compressible deep neural network"
_ICLR.cc/2020/Conference — Accept (Spotlight)_

### Official Review · AnonReviewer2 · 2019-10-23
**Official Blind Review #2**

**Rating:** 6

**Review:**

This paper provides generalization bound based on the compression arguments. A key contribution is that instead of bounding the
population risk for the compressed model, this paper manages to give bounds on the non-compressed network following a unified analysis framework. Also, this paper applies the unified framework to low-rank assumption on weigh matrices and covariance matrices.

Overall, I believe this paper should be accepted because of its contribution to our understanding of generalization theory. The central contribution here is using localized Rademacher complexity to bound the $L_2$ norm between original network and compressed network. However, I think there are several issues unclear and in need of clarification.

One thing I hope the authors could clarify is the novelty in the proof of Theorem 1, because it seems the techniques used here such as entropy integral and peeling are all well-known. It would be better if the authors could give a comparison between this paper and papers with similar techniques.

Another question is, in the statement of Theorem 1, the term $\sqrt{M \frac{2t}{n}}$ is marked as part of 'main term' and the term $C \dot r \sqrt{\frac{t}{n}}$ is marked as part of 'fast term'. I hope the authors could give a more detailed explanation of why $\dot r$ could be seen as a faster term than a constant, which seems not sensible to me.

Also, I hope the authors could explain why $\Phi(\sqrt{2(\hat r ^2 + r ^2 _*)})$ is a 'faster term', because it seems this term is not faster than $\sqrt{\frac{1}{n}}$.

The questions above essentially concern how sharp this general bound could be. I would appreciate it if the authors could give a thorough response to questions mentioned above. This can help me achieve a better understanding and a more precise evaluation on this paper.

**Experience Assessment:**

I have read many papers in this area.

**Review Assessment: Checking Correctness Of Derivations And Theory:**

I assessed the sensibility of the derivations and theory.

**Review Assessment: Checking Correctness Of Experiments:**

N/A

**Review Assessment: Thoroughness In Paper Reading:**

I read the paper at least twice and used my best judgement in assessing the paper.

---

> ### Author Response · Authors · 2019-11-14
> **Reply from authors**
>
> Thank you for your suggestive comments, which clarify our paper's contribution.
>
> > One thing I hope the authors could clarify is the novelty in the proof of Theorem 1, because it seems the techniques used here such as entropy integral and peeling are all well-known. It would be better if the authors could give a comparison between this paper and papers with similar techniques.
>
> In the literature, the local Rademacher complexity has been used to derive a fast leaning rate for a model the complexity of which is pre-determined. Howerver, the current setting is that the complexity of the trained model is more data-dependent, and the local Rademacher complexity is used to "bridge" the complexity of the trained network and the set of the small sized networks, G, while typical usage of the technique is directly bound the population excess risk. Hence, the usage of the local Rademacher complexity is quite different from the existing studies. In particular, we need to use the ratio type empirical process. Moreover, the bound is not restricted to the empirical risk minimization and it can be applied to any estimator as long as it produces a compressible network. We think our usage of the technique is interesting and this technique has not been employed in the literature of the generalization error analysis of overparameterized neural networks. Hence, although using the local Rademacher complexity might seem classical, we believe that it still provides an important insight to understand the generalization error analysis of deep learning.
>
> > Another question is, in the statement of Theorem 1, the term  is marked as part of 'main term' and the term  is marked as part of 'fast term'. I hope the authors could give a more detailed explanation of why \dot{r} could be seen as a faster term than a constant, which seems not sensible to me.
>
> If \hat{r} is fixed independent of the sample size, then \dot{r} is just a constant. However, by balancing the bias and variance terms, we may decrease \hat{r} as the sample size increases. Indeed, we took $\hat{r} =(\sum_l m_l/L)^{-\frac{1}{4/\beta + 2(1-1/2\alpha)}}$ in Theorem 4 which can be small if $m_l$ is relatively large (in particular, m_l increases as n goes up). However, we also realized that the terminology "fast term" would be confusing. We changed this to "bias term."
>
> > Also, I hope the authors could explain why $\Phi(\dot{r})$ is a 'faster term', because it seems this term is not faster than $\sqrt{1/n}$.
> Lemma 2 shows that the main term of $\Phi(r)$ is proportional to $O(r^{1-q}\sqrt{1/n})$ under some assumptions. Thus, if $\dot{r}$ is much smaller than 1, then $\Phi(\dot{r})$ can be smaller than \sqrt{1/n}. At least, we can show r^{*2} is o(1/\sqrt{n}) in a typical setting. This is why we used the terminology of "faster term."
>
> For the reasoning we have mentioned above, we employed the terminology "faster term." However, by balancing the bias and variance term, some part of the faster term becomes the same rate as the main term (this is mainly due to the term related to $\hat{r}$). We thought this terminology facilitates the understanding, but it could cause some confusion. Hence, we have changed the faster term to "bias term." We also modified the expositions after Theorem 1 so that there does not appear confusion.

---

### Official Review · AnonReviewer1 · 2019-10-24
**Official Blind Review #1**

**Rating:** 8

**Review:**

The paper presents novel theoretical results on generalization bounds via compression. Similar ideas in the last few years appeared, but only bounds on a compressed network were obtained. In contrast, the current submission gives a bound on the original (uncompressed) network in terms of the complexity of the compressed network class.

Overall, the paper seems to be well-written. I appreciate that the outlines of the proofs are included in the main text, which helps the reader follow the ideas. The result is novel and quite interesting. The new bounds seem to be still quite far from giving tight generalization theory, but I believe the paper provides some nice theoretical results for other researchers to improve upon. I think the paper could be improved immensely by some empirical analysis of the rank of compressed standard vision networks and rank of activation covariance matrices.  There are also some citation issues (see detailed comments below).

Citation issues:
In the introduction, paragraph 2, the authors cite Neyshabur et al. 2019 for the observation that networks generalize well despite being overparameterized. It seems like an odd choice. Why is Barttlet’s ‘99 paper [“Size of the weights…”]  not cited? Or at least Neyshabur et al. 2015?
Then the authors mention that classical learning theory cannot explain the phenomena mentioned above, and classical theory “.. suggests ” that overparameterized models cause overfitting…”. The authors need to be more precise and add citations (I am assuming that the authors are talking about VC bounds for worst-case ERM generalization).
In the third paragraph, where the authors talk about norm-based bounds being lose, it seems that Nagarajan and Kolter 2019 should be cited (not only at the end), as well as Dzigaite and Roy 2017 (they look into the looseness of path-norm and margin-based bounds).

Could the authors comment more on how the bound in Theorem 2 is superior to VC dimension bound and whether conditions under which the bound is tight are realistic for standard compressed vision networks. Having weight matrices to be close to rank 1 seems unrealistic.I would like to see some sort of empirical evidence if the authors believe that this is the case. And for larger ranks, the bound seems to be close to VC bound.

In general, I found the notation a bit hard to follow and had to constantly be looking through the paper to find the definitions of various quantities. Having three different r’s, multiple mu’s with dots, bars, stars, etc., was definitely confusing and required extra attention to detail.

Other minor comments:

In section 2, marginal distributions over x and y are introduced. Are those used in the main text?
Is that a definition of \mu with the dot on top in assumption 5, or is this mu with the dot defined earlier? Using notation := would make it clearer whether the quantity is being defined.
In Section 3, “The main difference from the…” paragraph, there is \Psi(\dot r) used. Where is that defined?

**Experience Assessment:**

I have published in this field for several years.

**Review Assessment: Checking Correctness Of Derivations And Theory:**

I assessed the sensibility of the derivations and theory.

**Review Assessment: Checking Correctness Of Experiments:**

N/A

**Review Assessment: Thoroughness In Paper Reading:**

N/A

---

> ### Author Response · Authors · 2019-11-14
> **Reply from authors**
>
> Thank you very much for your several comments. We have revised our paper according to your comments.
>
> > I appreciate that the outlines of the proofs are included in the main text, which helps the reader follow the ideas.
> Thank you very much for your suggestion. We will definitely add the outline of the proof in the final version.
>
> > I think the paper could be improved immensely by some empirical analysis of the rank of compressed standard vision networks and rank of activation covariance matrices.
> Thank you for pointing out this. We have added a numerical experiments about the eigenvalue distributions and intrinsic dimensionality of practically used VGG-19 network in Appendix D. We can see that the eigenvalues of the covariance and weight matrices decrease rapidly and the intrinsic dimensionality can be much smaller than the actual number of parameters. We also included a comparison with Arora et al. 2018. Our suggested quantity gives favorably tight evaluation compared with their numerical results.
>
> Citation issues:
> > In the introduction, paragraph 2, the authors cite Neyshabur et al. 2019 for the observation that networks generalize well despite being overparameterized. It seems like an odd choice.
> > Why is Barttlet’s ‘99 paper [“Size of the weights…”]  not cited? Or at least Neyshabur et al. 2015?
>
> Thank you very much for pointing out the citation issue. We cited Neyshabur et al. 2019 as a good pointer to the recent literature of generalization error analysis and numerical experiments on overparameterized networks. However, we agree with your opinion that the papers you mentioned should be cited. We have cited them in the revised version.
>
> > Then the authors mention that classical learning theory cannot explain the phenomena mentioned above, [...]
> > The authors need to be more precise and add citations (I am assuming that the authors are talking about VC bounds for worst-case ERM generalization).
>
> Yes, we intended that the "classical learning theory" is the VC-dimension type worst case analysis. The VC-dimension of networks with depth L and width W is lower bounded by L^2W^2, which yields O(\sqrt{L^2W^2/n}) of the generalization error bound (Harvey et al. 2017). We have modified this part as "well explained by a classical VC-dimension type theory (Harvey et al. 2017))" by citing the paper, Harvey et al. (2017).
>
> > In the third paragraph, where the authors talk about norm-based bounds being lose, it seems that Nagarajan and Kolter 2019 should be cited (not only at the end), as well as Dzigaite and Roy 2017 (they look into the looseness of path-norm and margin-based bounds).
>
> Thank you very much for the citation information. We could have missed some relevant papers, but we have cited them in the revised version.
>
> > Could the authors comment more on how the bound in Theorem 2 is superior to VC dimension bound and whether conditions under which the bound is tight are realistic for standard compressed vision networks.
>
> Our bound is always tighter than VC-dimension bound, but VC-dimension bound is recovered if we let \alpha and \beta = 0 small as an extreme case (this is not directly obtained by the presented theorems, but can be seen from the proof). As long as the singular value decay satisfies the assumptions, our bound can be tighter than VC-dimension bound (please remark that this does not necessarily imply the matrices are close to rank "1"). To show this is realistic, we have conducted numerical experiments (Appendix D). We can see that both of the weight matrices and the covariance matrices show rapid decrease of spectrum.
>
>
> > In general, I found the notation a bit hard to follow.
> Thank you very much for reading our paper in details. We will do our best to make the notation more concise in the final version.
>
> > Other minor comments:
> > In section 2, marginal distributions over x and y are introduced. Are those used in the main text?
> Thank you for pointing out this. They are used only in Assumptions 1 and 2 for clarifying the support of the distributions. We have moved the definition just before the assumptions.
>
> > Is that a definition of \mu with the dot on top in assumption 5, or is this mu with the dot defined earlier? Using notation := would make it clearer whether the quantity is being defined.
> Yes, you are absolutely correct. We have added ":="" in the right hand side.
>
> > In Section 3, “The main difference from the…” paragraph, there is \Psi(\dot r) used. Where is that defined?
> We appreciate your correction. This was typo. $\Psi(\dot r)$ is defined in Appendix A, but this was not defined before Section 3. We replaced this by $\dot{R}_{\dot{r}}(\widehat{\mathcal{F}} - \widehat{\mathcal{G}})$ in the revised version.

---

> > ### Comment · AnonReviewer1 · 2019-11-15
> > **Comments on the revised submission**
> >
> > Thank you for clarifying the notation and answering my questions. I found the new empirical evaluation of intrinsic dimensionality and comparison to Arora et al. bound to be a valuable contribution (Appendix D). However, I do believe that the presentation of the theoretical results and the notation in the main text could be made easier to follow still (I personally think that it is confusing to denote quite different quantities with the same letter but different subscripts or superscripts).
> >
> > Overall, I think the paper improved and therefore I increased my score to "accept".

---

### Official Review · AnonReviewer4 · 2019-11-03
**Official Blind Review #4**

**Rating:** 6

**Review:**

This paper obtains a compression-based generalization bound (Theorem 1) for the original network, while prior work gives bounds for the compressed network. The general bound given by Theorem 1 is further applied to networks with low-rank weight matrices (Theorem 2 and Corollary 1) or low-rank covariance matrices (Theorem 3 and 4). In some cases, the bound given by Theorem 1 for the original network could be better than the bound for the compressed network.

In terms of proof techniques, Lemma 2 is a general result to control the local Rademacher complexity using upper bounds on the covering numbers, which is interesting and could be useful in other problems.

On the other hand, there are two technical concerns.
(1) In eq. (5), the covering number of {\phi(f)-\phi(g)} is bounded by the covering number of {f-g}, which is not necessarily true. For example, in the 1-dimensional case, it is possible that f-g is always 1, while \phi(f)-\phi(g) is not a constant. This example might appear since f and g are not freely chosen from F and G; they further need to satisfy the condition that |f-g|_{L_2} is bounded by r. If the claim in eq. (5) is indeed true, a proof is needed.
(2) Despite the issue in (1), many bounds in the paper may actually be okay, since in the proofs the covering numbers of F (the original networks) are used (e.g., in eq. (6) and Lemma 2). Therefore it looks like the local Rademacher complexity of F can be controlled directly using Lemma 2. The question then is how compression helps in the analysis?

I hope the above points can be clarified, and I would like to participate in the discussion.

**Experience Assessment:**

I have read many papers in this area.

**Review Assessment: Checking Correctness Of Derivations And Theory:**

I assessed the sensibility of the derivations and theory.

**Review Assessment: Checking Correctness Of Experiments:**

N/A

**Review Assessment: Thoroughness In Paper Reading:**

I read the paper at least twice and used my best judgement in assessing the paper.

---

> ### Author Response · Authors · 2019-11-14
> **Reply from authors**
>
> Thank you for your suggestive comments.
>
> (1)
> Q: How to derive Eq. (5)?
> A: Thank you for clarifying the details. This is due to the Lipschitz continuity of the loss $\psi$. Indeed, we have that $|\psi(f(x)) - \psi(g(x))| \leq |f(x) - g(x)|$ which yields $\|\psi(f) - \psi(g)\|_n \leq \|f - g\|_n$. Then, we obtain Eq.(5). This is explained just after the equation as "where we used 1-Lipschitz continuity of the loss function $\psi$ in the ﬁrst inequality". However, we have added more detailed exposition to clarify this point in the revised version.
> As you pointed out, this is not true in general if it does not have Lipschitz continuity. The condition "|f-g|_{L_2} <= r" is not used to derive Eq.(5) itself, but only the Lipschitz continuity is used.
>
> (2)
> Q: it looks like the local Rademacher complexity of F can be controlled directly using Lemma 2. The question then is how compression helps in the analysis?
> A: Yes, you are right. The local Rademacher complexity of F appears only in the bias term. The variance term (the main term) is controlled by the (global) Rademacher complexity of G. If we don't consider compression, the main term (the global Rademacher complexity of G) must be replaced by that of "F" which is much larger than that of G. However, through compression, this becomes the complexity of G, which yields large improvement. We would like to notice that a naive analysis without local Rademacher complexity produces an additional term of a global Rademacher complexity of F to bound the bias term. Our technique resolves this issue by using the ratio type empirical process.

---

> > ### Comment · AnonReviewer4 · 2019-11-14
> > **How to prove the covering number inequality used in Eq. (5)?**
> >
> > Thanks for the response. I agree that the Lipschitz continuity of $\psi$ implies $|\psi(f(x))-\psi(g(x))|\le|f(x)-g(x)|$ and $\|\psi(f)-\psi(g)\|_n\le\|f-g\|_n$. However, to prove Eq. (5), it looks like the following inequality is used:
> > $$\mathcal{N}(\{\psi(f)-\psi(g)|f\in\widehat{\mathcal{F}},g\in\widehat{\mathcal{G}},\|f-g\|_{L_2}\le r\},\|\cdot\|_n,\epsilon)\le\mathcal{N}(\{f-g|f\in\widehat{\mathcal{F}},g\in\widehat{\mathcal{G}},\|f-g\|_{L_2}\le r\},\|\cdot\|_n,\epsilon).$$
> > I do not see how to prove it? It is not enough to only use $\|\psi(f)-\psi(g)\|_n\le\|f-g\|_n$; what we need to show should be something like given $h$ such that $\|(f-g)-h\|_n\le\epsilon$, it also holds that $\|(\psi(f)-\psi(g))-\phi(h)\|_n\le\epsilon$ for some transformation $\phi$.
> >
> > To put it simply, I agree that given a function class $\mathcal{F}$ and a Lipschitz function $\psi$, the covering number of $\psi(\mathcal{F})$ is bounded by the covering number of $\mathcal{F}$; however, I do not see why the above inequality is true, since we are considering $\psi(f)-\psi(g)$, not $\psi(f-g)$.
> >
> > In detail, in the original review I gave a special example where the above covering number inequality is not true. Consider the case $n=1$, and the set $A:=\{(z+1,z)|z\in[-1,+1]\}$. Then $\{x-y|(x,y)\in A\}$ only contains a single number $1$, and thus has covering number $1$. On the other hand, let $\psi$ denote the sigmoid function $e^x/(1+e^x)$ which is Lipschitz, then $\{\psi(x)-\psi(y)|(x,y)\in A\}=[\frac{1}{2}-\frac{1}{1+e},\frac{\sqrt{e}-1}{\sqrt{e}+1}]$, whose covering number is larger than $1$. This example is too special since in $A$, the two coordinates are not independent, and probably we can prove the above covering number inequality when $f$ and $g$ are freely chosen from $\widehat{\mathcal{F}}$ and $\widehat{\mathcal{G}}$; however they further need to satisfy the condition $\|f-g\|_{L_2}\le r$, which makes the situation more complicated.
> >
> > In addition, it looks to me that Eq. (5) is important to bound the bias term, which is after all the key term this paper tries to bound, as the title suggests "compression based bound for non-compressed networks". Therefore the discussion of Eq. (5) is not just a technical one, but could affect the big picture.

---

> > > ### Author Response · Authors · 2019-11-15
> > > **We have fixed the issue about Eq.(5)**
> > >
> > > Thank you for your thorough exposition.
> > > We have realized that you are absolutely correct.
> > > This issue can be easily fixed by replacing $\dot{R}_{r}(\widehat{\mathcal{F}} - \widehat{\mathcal{G}})$ with $\dot{R}_{r}(\psi(\widehat{\mathcal{F}}) - \psi(\widehat{\mathcal{G}}))$ as an upper bound of Eq.(5). This is further bounded by
> > > $$
> > > \dot{R}_{r}(\psi(\widehat{\mathcal{F}}) -\psi(\widehat{\mathcal{G}})) \leq
> > > \frac{C}{n}
> > > +
> > > C \mathrm{E}_{D_n}\left[ \int_{1/n}^{\hat{\gamma}_n} \sqrt{\frac{\log(N( \widehat{\mathcal{G}},\|\cdot\|_{n},\epsilon/2))}{n}} d \epsilon
> > > +
> > > \int_{1/n}^{\hat{\gamma}_n} \sqrt{\frac{\log(N( \widehat{\mathcal{G}},\|\cdot\|_{n},\epsilon/2))}{n}} d\epsilon \right].
> > > $$
> > > Please check Eq.(6) of the revised version. We also used this upper bound to bound $\dot{R}_{r}(\widehat{\mathcal{F}} - \widehat{\mathcal{G}})$ in the previous version and all the remaining arguments (Theorems 2,3 and 4) are derived from the the Dudley integral bound appearing in the right hand side instead of $\dot{R}_r$ itself. Therefore, this modification does not affect the remaining arguments. According to this modification, we fixed the main text and the proofs. They are just minor modifications.
> > >
> > > Although we used only the Dudley integral bound to show Theorems 2,3 and 4, we used the local Rademacher complexity $\dot{R}_{\dot{r}}(\widehat{\mathcal{F}} - \widehat{\mathcal{G}})$ in Theorem 1 to avoid a heavy notation related to the covering number appearing in the Dudley integral. Unexpectedly, this caused a mistake, but the $\dot{R}_{\dot{r}}$ term can be replaced by the Dudley integral anyway.
> > >
> > > Finally, we would like to remark that the local Rademacher complexity $\dot{R}_{r}(\widehat{\mathcal{F}} - \widehat{\mathcal{G}})$ is still required to bridge $\|f-g\|_n$ and $\|f-g\|_{L_2}$. Thus, it remains in the main text.
> > >
> > > We appreciate your insightful comment.

---

> > > > ### Comment · AnonReviewer4 · 2019-11-15
> > > > **Thanks for the response**
> > > >
> > > > It helps a lot!

---

### Author Response · Authors · 2019-11-14
**Revised version has been uploaded**

Dear reviewers,

Thank you very much for your insightful comments. We have revised our manuscript according to your comments. The main modifications are as follows:
1. We have added numerical evaluation in Appendix D. It evaluates the eigenvalue distribution of VGG-19 trained on CIFAR-10 and computed the intrinsic dimensionality of that.
2. We have added some missing important citations.
3. We have modified the intuitive explanations of the general bound (Theorem 1).

Sincerely yours,
Authors.

---

### Decision · Program_Chairs · 2019-12-19

**Decision:**

Accept (Spotlight)

**Comment:**

This paper has a few interesting contributions: (a) a bound for un-compressed networks in terms of the compressed network (this is in contrast to some prior work, which only gives bounds on the compressed network); (b) the use of local Rademacher complexity to try to squeeze as much as possible out of the connection; (c) an application of the bound to a specific interesting favorable condition, namely low-rank structure.

As a minor suggestion, I'd like to recommend that the authors go ahead and use their allowed 10th body page!